# Finding influential nodes for integration in brain networks using optimal percolation theory

Gino Del Ferraro[1], Andrea Moreno [2], Byungjoon Min[1,3], Flaviano Morone[1], Úrsula Pérez-Ramírez[4], Laura Pérez-Cervera[2], Lucas C. Parra [5], Andrei Holodny [6], Santiago Canals[2] & Hernán A. Makse[1]

Global integration of information in the brain results from complex interactions of segregated brain networks. Identifying the most influential neuronal populations that efficiently bind these networks is a fundamental problem of systems neuroscience. Here, we apply optimal percolation theory and pharmacogenetic interventions in vivo to predict and subsequently target nodes that are essential for global integration of a memory network in rodents. The theory predicts that integration in the memory network is mediated by a set of low-degree nodes located in the nucleus accumbens. This result is confirmed with pharmacogenetic inactivation of the nucleus accumbens, which eliminates the formation of the memory network, while inactivations of other brain areas leave the network intact. Thus, optimal percolation theory predicts essential nodes in brain networks. This could be used to identify targets of interventions to modulate brain function.

[1] Levich Institute and Physics Department, City College of New York, New York, NY 10031, USA. [2] Instituto de Neurociencias, CSIC and UMH, 03550 San Juan de Alicante, Spain. [3] Department of Physics, Chungbuk National University, Cheongju, Chungbuk 28644, Korea. [4] Center for Biomaterials and Tissue Engineering, UPV, Valencia, Spain. [5] Biomedical Engineering, City College of New York, New York, NY 10031, USA. [6] Department of Radiology, Memorial Sloan Kettering Cancer Center, New York, NY 10065, USA. These authors contributed equally: Gino Del Ferraro, Andrea Moreno. Correspondence and requests for materials should be addressed to S.C. (email: scanals@umh.es) or to H.A.M. (email: hmakse@lev.ccny.cuny.edu)

A fundamental question in systems neuroscience is how the brain integrates distributed and specialized networks into a coherent information processing system[1,2]. Brain networks are considered integrated when they exhibit long-range correlated activity over distributed areas in the brain[2–6]. Correlation of brain activity is typically measured using functional magnetic resonance imaging (fMRI), and the correlation structure is often referred to as "functional connectivity"[2–6].

Current network theory applied to such brain networks suggests that integration of specialized modules in the brain is facilitated by a set of essential nodes[2–4,7,8]. Perturbations in such essential nodes are therefore expected to lead to large disturbances in functional connectivity affecting global integration[2,5,8]. A number of neurological and psychiatric disorders have been attributed to disruption in the functional connectivity in the brain[5,9] and many of the alterations associated with brain disorders are likely concentrated on essential nodes[10–13]. Thus, identifying these essential nodes is a key step toward understanding information processing in brain circuits, and may help in the design of targeted interventions to restore or compensate dysfunctional correlation patterns in disease states of the brain[9].

There are several studies that have used network centrality measures to identify the essential nodes in brain networks[3–6,8,9,14–17]. These measures include the hubs (nodes with many connections), betweenness centrality (BC)[18], closeness centrality (CC)[19], eigenvector centrality (EC)[20,21], the $k$-core[22,23], and collective influence (CI) centrality which uses optimal percolation theory[24] to identify essential nodes[8] (see [16,25] for a review).

These centrality measures can be used as a ranking to determine the most influential nodes in brain networks, and nodes with the highest ranking are considered to be the "essential" nodes for integration. While each centrality provides a different aspect of influence[16], a common prediction of all measures is that when the essential nodes are inactivated in a targeted intervention, integration in the overall network is largely prevented[2,5,8]. That is, when inactivated, nodes with the highest rank lead to the largest damage to the long-range correlations. Thus, the optimal centrality measure would be the one which prevents integration of the network by inactivating the fewest number of nodes[24,26]. The minimal set of nodes that upon inactivation destroy the integration of the network is obtained by mapping the problem to optimal percolation[24]. Finding this minimal set of essential nodes is an NP-hard problem in general[26]. Yet, it can be approximately solved with an efficient algorithm called Collective Influence (CI) assuming sparse network connectivity[8,24].

Some of the centrality measures have been studied using analytical and numerical methods, and have been associated with different clinical phenotypes[5,9,16]. However, their importance for brain integration has not been directly tested experimentally with prospective interventions. The effects of removing a node from a network has been studied with simulations, both for human and animal brain networks[11,27,28], but direct in vivo validations are rare. Thus, there is no well-grounded approach to predict which nodes are essential for brain integration.

Here, we address this problem empirically in an in vivo rodent preparation. We experimentally generate a network of long-range functional connections between diverse brain areas. Specifically, we induce synaptic long-term potentiation (LTP) in the rat dentate gyrus[29], which results in correlated evoked fMRI activity in brain areas that are involved during memory encoding and consolidation. These include the hippocampus (HC), the prefrontal cortex (PFC), and the nucleus accumbens (NAc)[30]. The key question is this: Which are the essential nodes in this memory network that are necessary for these long-range functional interactions to form. We first identify the nodes that maximally disrupt the integrated memory network by systematic inactivation of essential nodes identified following the different centrality criteria. We find that centralities fall into two classes: hub-centralities (degree, $k$-core, EC) which only identify the hubs at the stimulation site (the HC), and integrative centralities (CI and BC) which identify "weak nodes", i.e., low-degree yet highly influential nodes for brain integration, notably, in the NAc. Using pharmacogenetic inactivation[31], we validate in vivo the theoretical prediction, namely, that weak nodes in the shell of the NAc are essential for the integration into a larger memory network. These experimental results confirm the importance of going beyond the direct connection of hubs and instead considering the CI of nodes on network integration[24].

## Results

**Overall approach**. Our combined experimental and modeling approach takes the following steps: First, induce a functional network in vivo using synaptic LTP in the rat HC. Second, model this functional brain network as the result of pairwise interactions in a sparse brain network. Third, identify and compare the essential integrators using various centrality criteria based on the topology of the brain network. Finally, inhibit the predicted essential and non-essential nodes in the in vivo preparation and test whether network integration is prevented only for essential nodes, as predicted by the theory. In the following, we elaborate on each of these steps.

**Experimentally coupling functional networks in vivo**. LTP of synaptic connections is considered the cellular basis of learning and memory[29]. Combined fMRI and electrophysiological experiments have demonstrated that LTP induction in the perforant pathway, the major entorhinal cortex input to the dentate gyrus, causes a lasting increase of fMRI activity in distant brain areas such as neocortical and mesolimbic sites (PFC and NAc)[30]. This result suggests that the impact of local synaptic plasticity is not restricted to the synaptic relay at which it is induced, as it is so usually studied, but can facilitate long-range propagation of activity more broadly into a network formed by the different activated areas in the brain. While this network formation is known to depend on the activation of $N$-methyl-D-aspartate (NMDA) receptors[30], the precise mechanisms and relative importance of the different structures to its formation are not known[32]. Thus, this LTP paradigm represents an ideal system to investigate the essential nodes for long-range integration.

We follow a well-characterized protocol to induce LTP (details of experiments in Fig. 1a–d and Supplementary Note 2) and apply high-frequency pulsed stimulation (250 Hz) of the perforant pathway of the HC in six rats. We apply low-frequency stimulation (10 Hz) before (PRE) and 3 h after (POST) LTP induction, to evoke activity in the hippocampal formation while concurrently performing fMRI. Low-frequency stimulation does not affect synaptic efficacy but does allow us to measure activated brain areas with fMRI (e.g., Fig. 1c shows response to stimulation relative to baseline at $p < 0.001$, corrected). We verify that synaptic potentiation is induced by the high-frequency stimulation by measuring the concomitant electrophysiological recordings from the dentate gyrus as shown in Fig. 1b, e, f.

LTP induction results in the propagation of evoked fMRI activity to a long-range functional network beyond the site of low-frequency stimulation (ipsilateral HC). Activations after LTP induction (POST) are reported in Fig. 1g for a single animal, and in Supplementary Fig. 1a for the average over six animals. Compared to the baseline activation (PRE), we see enhanced bilateral fMRI activation of the HC, and activation in frontal and prefrontal neocortical regions (PFC), as well as the NAc (see

Fig. 1h, i for group results and statistics; see also Supplementary Note 2). Conversely, low-frequency stimulation of the perforant pathway before LTP induction produces no fMRI activity in the PFC nor in the NAc (Fig. 1i).

**Generate a brain network model.** The voxels with significant fMRI activation (due to the low-frequency probe after LTP induction) form the nodes of the network model (see Supplementary Note 3 for details). We focus on evoked activity as we are interested in propagating functional activity in the memory network, rather than spontaneous resting state activity, which will be discussed further below (Section 2). The fMRI signal of the activated voxels is used to compute a functional connectivity matrix, i.e., pairwise correlations between voxels, separately for each animal. To build the computational model of the functional network, we proceed in two steps. First, we identify the clusters of nodes associated with different brain areas, and then we determine the "connectivity" between nodes.

It is well established that the functional connectivity matrix exhibits a modular structure, with modules (or clusters of nodes) typically associated with different anatomical brain areas[33]. To identify these modules, we follow standard procedures[4], namely, the functional connectivity matrix is thresholded and a 'community detection' algorithm is applied on this binarized

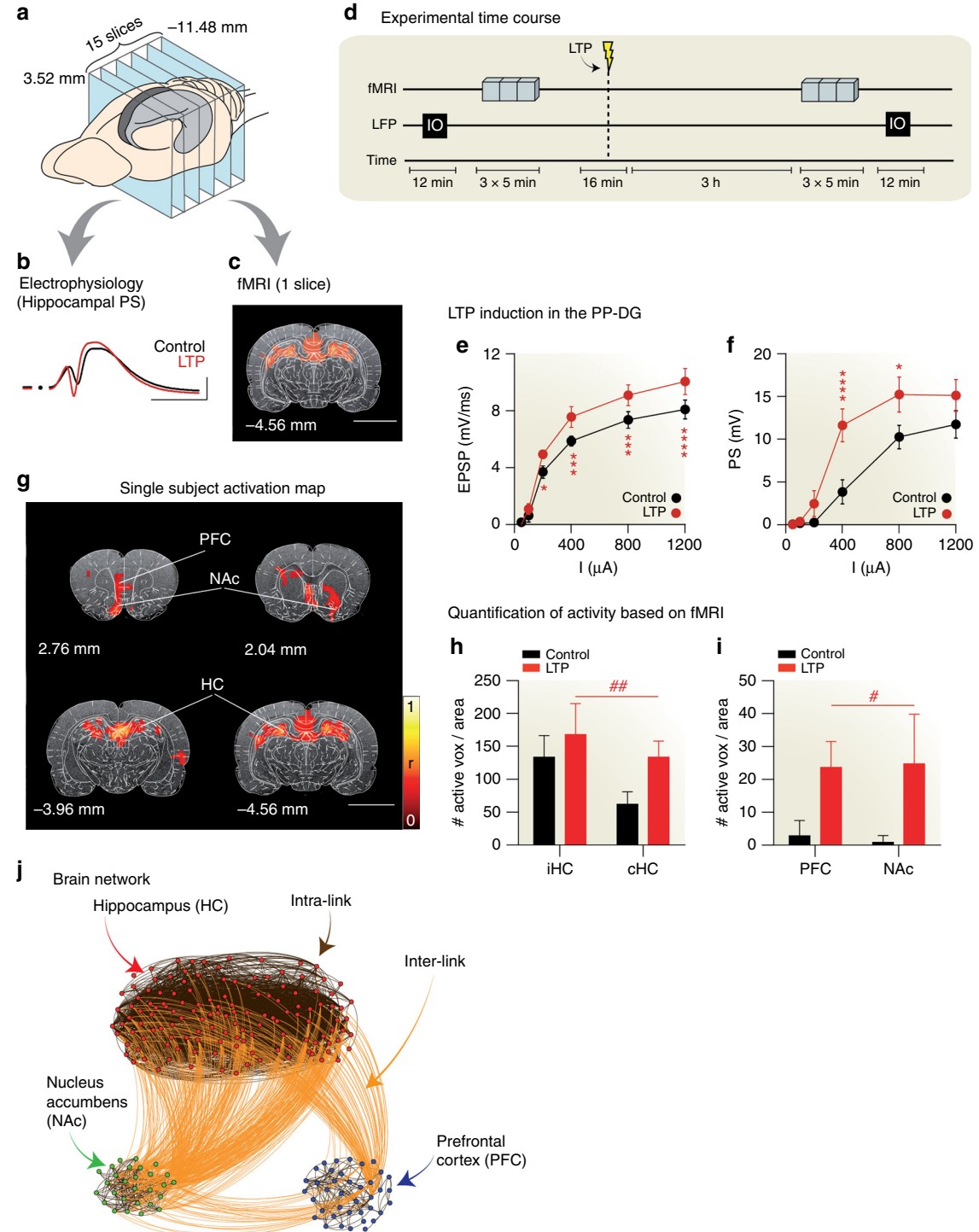

**a** 15 slices −11.48 mm 3.52 mm

**b** Electrophysiology (Hippocampal PS) — Control / LTP

**c** fMRI (1 slice) −4.56 mm

**d** Experimental time course
fMRI — LFP — IO — Time
12 min | 3 × 5 min | 16 min | 3 h | 3 × 5 min | 12 min
LTP

**e** LTP induction in the PP-DG — EPSP (mV/ms) vs I (μA) — Control / LTP

**f** PS (mV) vs I (μA) — Control / LTP

**g** Single subject activation map — PFC, NAc, HC — 2.76 mm, 2.04 mm, −3.96 mm, −4.56 mm — r: 0 to 1

Quantification of activity based on fMRI

**h** # active vox / area — iHC, cHC — Control / LTP — ##

**i** # active vox / area — PFC, NAc — Control / LTP — #

**j** Brain network — Hippocampus (HC), Intra-link, Inter-link, Nucleus accumbens (NAc), Prefrontal cortex (PFC)

matrix[7,34–36]. We also register each brain to a standard anatomical atlas (Paxinos and Watson rat brain atlas[37]). With this approach, we identified in each of the six animals three dominant clusters of nodes (voxels), which overlap well with the anatomical location of the HC, the PFC, or the NAc (Supplementary Fig. 1b).

The conventional approach to generating a "connectivity" matrix in brain networks models is to directly threshold the fMRI correlation matrix[4]. However, correlations do not only arise because two nodes exchange information or are directly linked, but may arise due to common covariates. Furthermore, "spurious connections" may result from a small sample size of the time series used to compute correlations. To minimize the effects of indirect covariation and sampling noise, we use a well-established statistical inference method[38]. This method models the observed correlations as the result of direct pairwise interactions, and imposes a penalty to avoid negligible interactions. By varying a penalization parameter, this widely used approach tunes the sparsity of the network. As with the direct thresholding of the correlation matrix[4,7,39], there are various ways to select this penalization parameter. We are interested in the formation of a connected brain network, where the different brain areas are linked with each other. Mathematically, this corresponds to the emergence of the "giant connected component" covering the entire network, i.e., all the nodes are connected through a path[7,8]. We selected the penalization parameter that results in the sparsest network which still exhibits a giant connected component (see also Supplementary Note 3 for details).

In the following, the connections within each cluster are referred to as *intra-links*, descriptive of short-range interactions within nodes in the same sub-network[40]. Connections between nodes belonging to different clusters are named *inter-links*, or *weak-links*[7], reflecting the long-range interactions between different sub-networks. Inter-links between the HC, NAc, and PFC bind these networks into a unified brain network as seen in Fig. 1j for a typical rat (inter- and intra-links shown in orange and black, respectively)[7,8,41]. Once the network model has been constructed, we proceed to identify the essential nodes for integration.

**Identifying essential integrators in the brain network model.** We define global integration as the formation of the largest connected component of nodes in the network—the "giant connected component" $G$. This is the graph that connects the largest numbers of nodes through a path (highlighted in yellow in Fig. 2a; see Supplementary Note 3). The emergence of such a giant component is an important concept in percolation theory, which studies the behavior of clusters in networks as a function of

a thresholding parameter of the graph[42,43]. The essential integrators of the brain network are then the optimal set (minimal number) of nodes that, upon inactivation, lead to a disintegration of the giant component into smaller disconnected clusters. This is the problem of optimal percolation, which attempt to find such a minimal set of essential nodes or influencers[8,24]. Therefore, we search for the essential nodes by systematic, numerical inactivation of nodes predicted by optimal percolation theory, while we monitor the size of the giant component.

Inactivation proceeds in rank-order according to different centralities. We first apply the hub centrality and thus sort the nodes by their degree. While the hub-centrality is not optimal, it is interesting to see how the hubs rank in terms of network integration, since they have been identified as central to integration in previous studies. As it is customary in network theory[8,14,24,42,43], we quantify the damage made to the integration of the brain network by measuring the size of the largest connected component $G(q)$ after we remove a fraction $q$ of nodes, whereby nodes are removed in the order of degree from high to low. Figure 2c shows $G(q)$ under inactivation of a fraction of $q$ hubs (mostly HC nodes in red). The curve indicates that the inactivation of hubs does not propagate the damage to the rest of the network. That is, removal of 20% of hubs reduces the size of $G$ by the same amount to 80% of its original value for this representative animal. Further, almost all the hubs are located in the dentate gyrus of the HC. The hub map averaged over six animals which plots the density of essential hubs in the brain, that is, those hubs that create the largest damage upon inactivation (calculated in Supplementary Note 4), is shown in Fig. 2g and confirms that most of the essential hubs are located at the site of LTP induction in the dentate gyrus. This is not surprising since we stimulate its major input (the perforant pathway) to induce the functional brain network. Inactivating the largest hubs in the dentate gyrus experimentally would trivially disrupt the network formation by directly preventing its local activation, rather than by breaking the integration of the network. Thus, these top hubs are trivial influencers.

To find essential nodes beyond the hubs at the HC, we follow optimal percolation to estimate the minimal set of essential nodes[8,24] by ranking the nodes according to the CI algorithm[8]. We find that the ranking following the CI centrality requires the smallest number of inactivated nodes to break up the giant component since CI arises from a maximization of the damage done to the giant component[8,24]. The CI centrality is defined by Eq. (2) in Supplementary Note 1 and quantifies the influence of a node not only by its degree, but also by the degree of nodes located in spheres of influence of size $\ell$—we refer to this as the sphere of influence Ball($i, \ell$) of radius $\ell$. Thus, CI can identify also

**Fig. 1** Experimental protocol and generation of brain network. **a** Schematic representation of the imaging planes (blue). The hippocampus (HC) is highlighted in gray. Numbers indicate $z$ coordinate in mm from bregma. **b** Representative evoked population spike (PS) in the dentate gyrus before (black) and after (red) LTP induction. **c** Representative fMRI maps across the HC during perforant path stimulation overlaid on an anatomical T2-weighted image with atlas parcellations (see Supplementary Note 2). Color indicates significant correlation ($p < 0.005$ corrected). **d** Time course of the experiment. Input/output (I/O) response curves are recorded in the local-field potentials (LFP). fMRI signals are collected during low-frequency (10 Hz) test stimulations before and 3 h after LTP induction. **e** Field excitatory postsynaptic potential (EPSP) slope and, **f** population spike (PS) amplitude before (black) and after (red) LTP. Two-way repeated measures ANOVA ($n = 5$, $\alpha = 0.05$) reveals significant effects of LTP in both measures ($F_{1,24} = 27.82$, $p < 0.0001$, and $F_{1,24} = 59.89$; $p < 0.0001$ for PS and EPSP, respectively). Mean ± SEM. Post-hoc Bonferroni: *$p < 0.1$; **$p < 0.01$; ***$p < 0.001$; ****$p < 0.0001$ **g** Representative fMRI maps in one animal after LTP induction. Color code as in (**c**) ($p < 0.005$; see Supplementary Note Fig. 1 for group activation maps and Supplementary Note 2 for details). Size bar corresponds to 0.5 mm. **h, i** Number of active voxels per selected region in control (black) and LTP (red) conditions in hippocampal (**h**) and extra-hippocampal areas (**i**). The stimulated region is the ipsilateral hippocampus (iHC); two-way repeated-measures ANOVA ($n = 7$, $\alpha = 0.05$) reveals significant effects for LTP in hippocampal ($F_{1,12} = 15.72$, ##$p = 0.0019$) and extrahippocampal regions ($F_{1,12} = 7.426$, #$p = 0.0184$), with no interaction between regions ($F_{1,12} = 0.00242$, $p = 0.9616$ and $F_{1,12} = 1.518$, $p = 0.2415$ for hippocampal and extra-hippocampal regions, respectively). Mean ± SEM. **j** Brain network formed by the HC, NAc, and PFC for the animal in (**g**). The brain network is formed by intra-network interactions and inter-network interactions inferred from fMRI correlation data (Supplementary Note 3)

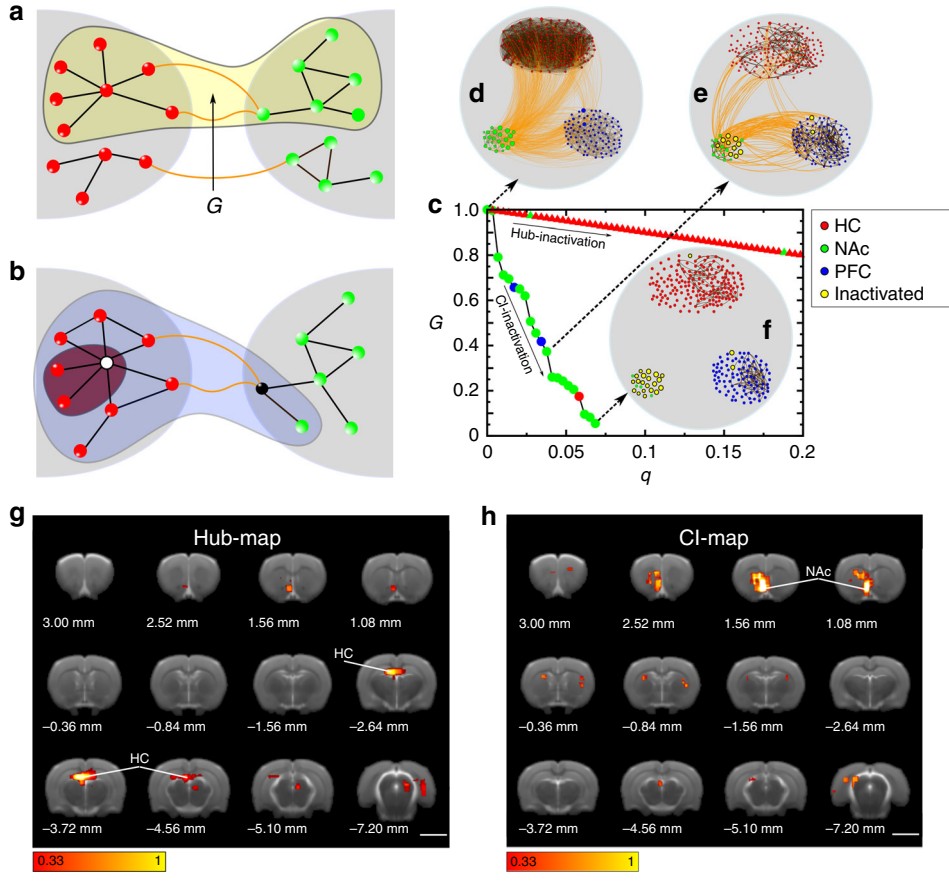

**Fig. 2** Hub and Collective Influence map. **a** The giant (largest) connected component $G$ (yellow) captures the integration of two modules into a brain network. **b** Influence of a hub and a CI node. Inactivation of the hub (white node) produces less damage to brain integration, measured by the size reduction of $G$, then the inactivation of the CI node (black). **c** Relative size of $G$ as a function of the fraction of inactivated nodes, $q$. Two strategies are shown for choosing the essential nodes in a representative animal: Hub inactivation (triangles) and CI inactivation (circles). Nodes are removed one by one according to their degree or CI-score, respectively, from high to low. Colors refer to the nodes module (HC, NAc, or PFC, see legend). Most hubs (red symbols) are located in HC, yet, they are not essential for integration: their removal makes minimal damage to $G$. On the contrary, by inactivating 7% of high CI nodes, $G$ collapses to almost zero. Most CI nodes are in the NAc (green symbols). **d** Representative brain network as in (**c**), displaying the PFC–HC–NAc networks. The size of each node is proportional to the CI score. **e** We inactivate the top 3% of high CI nodes (yellow circles) and $G$ is drastically reduced to less than 40% of its original value. These top CI nodes are all in the NAc except for two nodes in the PFC. **f** Further inactivating up to 7% of the high CI nodes prevents integration of $G$. Yellow circles indicate the essential nodes, located mostly in the NAc shell. **g** Average hub map indicating top hub nodes over six animals. Yellow/white areas correspond to top essential nodes all located in the HC since this is the area of LTP induction. Color bar represents the average rank (Supplementary Eq. (8)). **h** Average CI map indicating top CI nodes over six animals, most CI nodes result in the NAc and are generally not hubs. Color bar is defined in Supplementary Eq. (8), the size bar corresponds to 0.5 mm

low-degree nodes as influential as long as they are surrounded by high-degree nodes in their spheres of influence.

As shown in the particular animal in Fig. 2c, the giant component $G(q)$ quickly disintegrates when removing the top CI nodes (mostly NAc nodes in green). This result is consistent across all six animals (Supplementary Fig. 3). In clear contrast to the results obtained for hub-nodes, Fig. 2c shows that the removal of a very small fraction of top CI nodes (~7% of the total) is sufficient to reduce the giant component to 5% of its original size. Crucially, most of the nodes in this influential set are located in the NAc as shown in the sequence of network inactivation for this particular animal in Fig. 2d–f. Figure 2h shows the CI-map averaged over six animals, indicating that nodes essential to brain integration are located in the NAc according to the CI algorithm. This anatomical location is not predicted by conventional hub centrality since nodes in the NAc do not appear among the top hubs (Fig. 2g).

To illustrate the different network properties captured by hubs and CI centralities, consider Fig. 2b. Removing the node with the largest CI (depicted in black) results in large damage to the giant

connected component (shaded in blue). Removing the largest hub (depicted in white) causes relatively less damage (shaded in red). Thus, the different nodes predicted by the hub and CI maps are the result of long-range influence encoded in the CI measure, which is not captured by the local measure of degree. We note that the CI centrality includes the hub centrality as the zero-order approximation when we take a sphere of influence of zero radius, $\ell = 0$ in Supplementary Eq. (2). In this case, the influence centrality of Eq. (2) measures the number of connections of each node. When $\ell \geq 1$, CI captures effects emerging from the long-range structure.

The anatomical localization of essential nodes predicted by the other centrality measures is shown in Fig. 3. A detailed definition of these centrality measures is provided in the Supplementary Note 1. BC (BC-map, Fig. 3a) shares with the CI centrality (CI-map, Fig. 2h) a similar location of essential nodes in the brain, showing that the most influential nodes are located in the NAc shell. This indicates that the influential nodes are also bridge nodes captured by the BC.

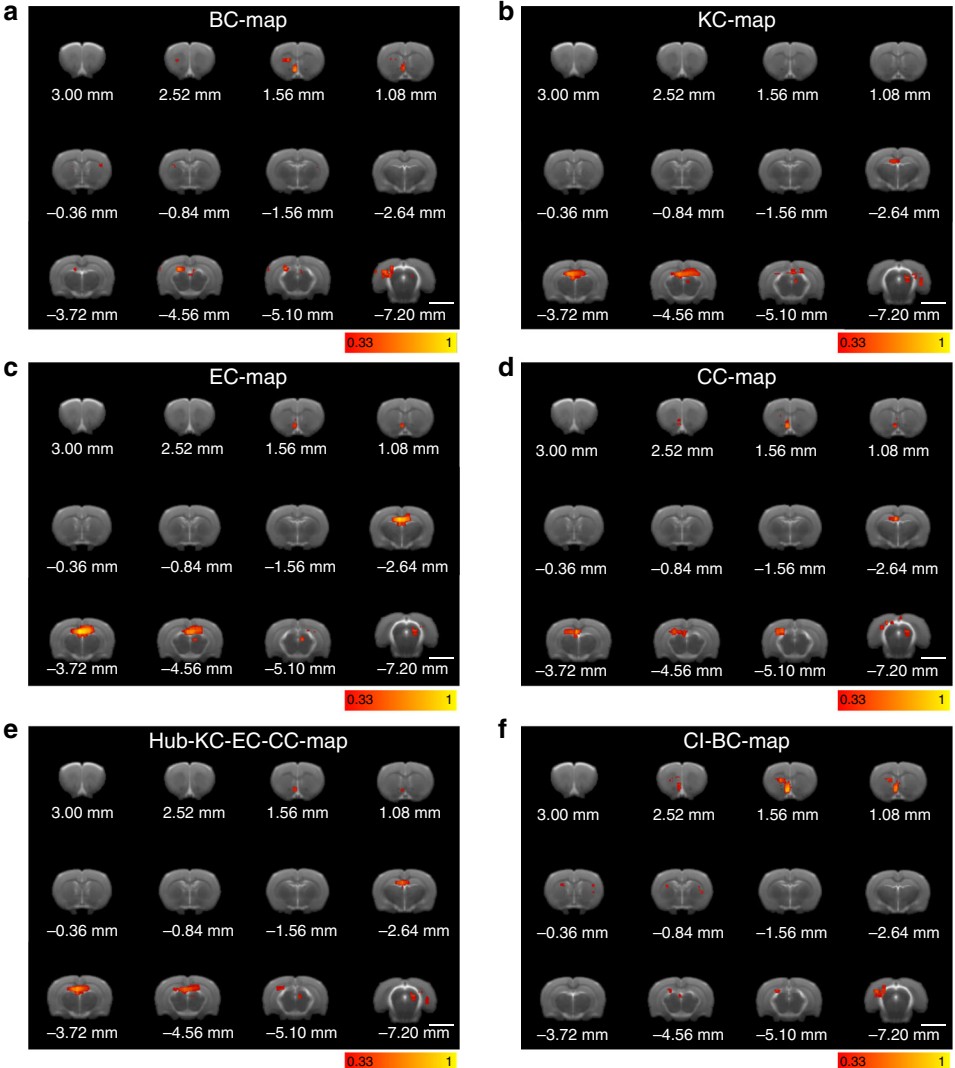

**Fig. 3** Maps of essential nodes. Average map of influencers for the different centralities according to **a** betweenness centrality, **b** $k$-core centrality, **c** eigenvector centrality, and **d** closeness centrality. The maps are averaged over the six rats and the color bars are calculated according to the rank defined by Supplementary Note 4, Eq. (8). Yellow/white colors indicate the top influencers according to each centrality. According to these results, the centralities are then divided into **e** hub-centric centralities dominated by the hubs and identifying the hubs in the HC and **f** integrative centralities dominated by the weak nodes and identifying the low-degree nodes in the shell part of the NAc. The size bar in each panel corresponds to 0.5 mm

In contrast, the NAc does not appear with high $k$-core centrality[22] (KC-map, Fig. 3b), which shows a distribution of essential nodes comparable to the hub map. This indicates that the nodes at the inner $k$-core of the network are correlated with their degree as expected by its definition. The EC (EC-map, Fig. 3c) also shows essential nodes mainly located in the HC, as expected since the eigenvectors of the adjacency matrix are highly localized by the hubs as shown in[44]. Finally, the CC (CC-map, Fig. 3d) shows essential nodes for integration in the HC and in the NAc to a lesser extent.

These results unveil a pattern in which centrality measures dominated by local degree (hubs, $k$-core, EC) tend to identify essential nodes in the hubs of the HC, since nodes with high degree are mostly located in the HC region. These nodes, in the present experiment, are trivially associated to the primary location of stimulation, while centrality measurements that capture long-range influence provide a non-trivial result highlighting the strength of the low-degree nodes at the NAc. The role of the NAc, thus, is analogous to a fundamental notion of sociology termed by

Granovetter as "the strength of weak ties"[7,45], according to which a weak tie (in our case a weak node, i.e., low degree, in the NAc) becomes a crucial bridge (a shortcut) between the densely knit clumps of close friends (the HC, NAc, and PFC). The average map of these two categories is shown in Fig. 3e (hub centric: hub-KC-EC-CC-map) and Fig. 3f (weak-node centric: CI-BC-map). In the Supplementary Note 7, we present the degree distribution of the CI nodes, across animals, and compare it with the distribution of the hubs. Supplementary Fig. 6 illustrates that most of the top CI nodes are low-degree nodes.

Overall, this comprehensive network analysis indicates that the integration among HC, NAc, and PFC triggered by LTP induction critically depends on the NAc, and not only on the largest network hubs at the activation site (HC), a fact that had not previously been recognized. The theory based on weak-node centralities predicts that the NAc is strategically located in the memory network, so that inactivating a small number of its nodes is sufficient to have the largest impact on the global connectivity; a falsifiable prediction that we test next.

**Targeted inactivation in-vivo in the real brain network**. In order to test these predictions, we repeat the LTP experiment in an additional five animals, while inhibiting the activity in the NAc region. The network module identified by the anatomic region in the NAc contains 33 nodes in a typical rat, corresponding to a 33 mm$^3$ volume. This activated module includes the NAc core and shell (which occupies approximately 10 mm$^3$ in the adult rat) as well as other areas surrounding the NAc. The theoretical prediction of CI identifies the top influencer around coordinate 2.5 anterior and 1.3 mm lateral from bregma and 7.0 mm ventral from the cortex surface, in Paxinos and Watson rat brain atlas space[37]. This location corresponds to a single node in the anterior half of the NAc shell. The pharmacogenetic intervention infects an approximate volume of 1 mm$^3$, thus silencing a volume corresponding approximately to one to two nodes (voxel volume) in the brain network structure, which allows specific testing of the analytical prediction.

We use adenoassociated viruses (AAV) to direct the expression of Designer Receptors Exclusively Activated by Designer Drugs (DREADDs)[31] into the particular targeted area of the NAc shell predicted as the top CI node. More specifically, we use the inhibitory version Gi-DREADD (hM4Di) which, under intraperitoneal administration of the otherwise inert ligand clozapine-N-oxide (CNO), activates the receptor inducing neuronal silencing and blocking the targeted high-CI node in the NAc shell. With this experimental design, we acquire fMRI data before and after administration of CNO, that is, in presence or absence, respectively, of a functional high-CI node located in the NAc shell of the network.

We favor the pharmacogenetic approach in this experiment over an optogenetic strategy because it avoids implanting bilateral cannula and optic fibers across frontal and/or prefrontal cortical regions from which we collect and analyze fMRI signals. We microinject the viruses bilaterally into the NAc and wait for 4 to 6 weeks to allow strong expression of the construct (see Fig. 4a, b and Supplementary Note 8). Two animals presented infection at neocortical regions due to leak of viral particles during the injection procedure and are not considered in further fMRI analysis. Histological verification demonstrates that viral expression is restricted to approximately a voxel in the shell part of the NAc (Fig. 4b). This subregional specificity is most likely produced by the virus serotype used (AAV5) and gives us the opportunity to selectively silence nodes in the NAc region receiving most HC input[46].

Before LTP induction, we perform a control experiment to inactivate the NAc shell. Comparing before and after CNO administration, (+) and (−) respectively, we find a comparable fMRI response to low-frequency stimulation in the HC: Both the fMRI activation maps (Fig. 4e, g) and the amplitude of the fMRI signals averaged across animals (Fig. 4f, h) are unchanged, demonstrating that the baseline fMRI response in the HC is not altered by NAc shell inactivation. Therefore, the input necessary to drive the formation of the memory network is preserved and can be used to experimentally test the theoretical predictions.

Using the same animals, we induce LTP in the perforant pathway as before but, this time, under inactivation of the NAc shell ((+) CNO). Figure 4i, j shows that, as predicted by the theory, the formation of the long-range network involving HC, PFC, and NAc is completely prevented, yet LTP induction still produces the expected potentiation of the intra-hippocampal bilateral activation (compare Figs. 4g, h and 4i, j). Remarkably, long-range inter-network links from the HC to the PFC are not formed (Fig. 4i, j), even though these sub-networks are not directly inactivated.

For comparison, the result of LTP induction in animals with a fully active NAc (animals without DREADD expression, (−)

AAV) is shown in Fig. 4c (fMRI activation map) demonstrating ipsilateral and contralateral HC activation together with PFC and NAc in response to the perforant pathway stimulation (averaged fMRI signal in Fig. 4d). These results demonstrate that inactivation of the highest CI node in the NAc shell disrupts the formation of the memory network by selectively blocking the formation of LTP-dependent connections to neocortical structures, but not the local potentiation of hippocampal synapses.

**Control experiments: in-vivo inactivation of brain regions predicted to have no effect**. To further validate these results, we perform a series of in vivo inactivation experiments targeting nodes which, based on our model predictions, should have no major effect on the long-range functional network.

We start with the inactivation of a node in the primary somatosensory cortex (S1), a brain region outside of HC-PFC-NAc functional network. Inactivation is first performed using DREADDs as before, with virus injection targeting the S1 region (Fig. 5a, see Supplementary Note 8 for details). As shown by the activation maps and fMRI signals in Fig. 5b, c, S1 inactivation does not prevent the LTP-induced activation of the HC-PFC-NAc network. Furthermore, in an additional group of animals we increased the strength of inactivation in S1 cortex by infusing 0.5 μL of tetrodotoxin (TTX, 100 μM) at the same stereotaxic coordinates (Fig. 5d). TTX is a sodium channel blocker that completely blocks neuronal firing at these concentrations (see Supplementary Note 9 for further details). Still, Fig. 5e, f demonstrates HC-PFC-NAc network formation upon LTP induction in these conditions.

Inactivation of the HC ipsilateral to the stimulation site would trivially eliminate the long-range network preventing its initial activation. We therefore tested whether inactivation of the contralateral HC nodes, identified by our model as non-essential nodes for global integration, would preserve network formation. As for S1 cortex, we used DREADDs (Fig. 5g) and TTX (Fig. 5j) in separate experiments to assure strong and wide inactivation of the contralateral HC (see Supplementary Notes 8 and 9, for details). The results with both manipulations verify our model prediction by showing successful LTP-induced formation of a long-range HC-PFC-NAc network under contralateral HC inactivation (Fig. 5h, i, k, l). Note that TTX injection prevents the activation of the complete contralateral HC, involving a large number of network nodes but nonetheless, the long-range network is preserved.

In our final control experiment, we targeted the DREADD inactivation to the anterior part of the PFC (Fig. 5m), a central part of the long-range network for which our model predicts low impact on global integration. TTX is not used for this target because the close proximity of the NAc and the diffusion of the TTX solution after injection cannot exclude direct inactivation of the NAc (and vice versa). However, the pharmacogenetic manipulation was enough to inactivate the PFC as demonstrated in the fMRI activation map and corresponding BOLD signals (Fig. 5n, o). Most importantly, under PFC inactivation, LTP successfully recruits the long-range HC-NAc network.

Between-groups statistical comparison (Fig. 6, see caption for statistics) demonstrates that only NAc inactivation promotes the complete disintegration of the LTP-induced HC-PFC-NAc network, while PFC targeting only produces the expected inactivation of the PFC and control S1, and contralateral HC inactivations preserve the complete long-range integrated network. Overall, these results lend strong support to the predictive validity of the model and the key role of the NAc in the LTP-induced long-range functional network.

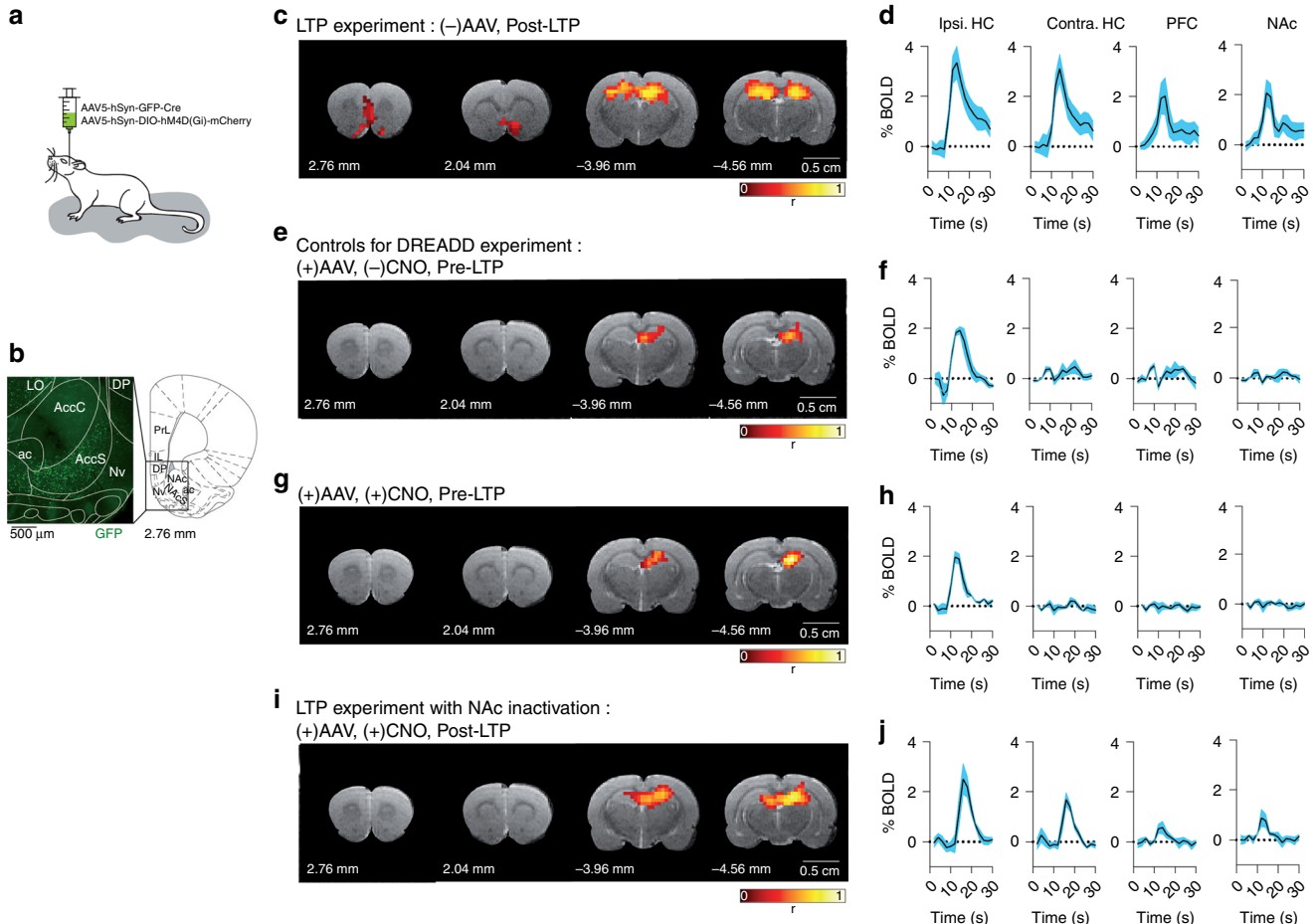

**Fig. 4** Experimental test of essential nodes. **a** The inhibitory version of DREADD receptors (hM4Di) is expressed in the NAc shell using a combination of two adenoassociated viruses (AAVs) injected stereotactically in the region as indicated (see Supplementary Note 8 for details). **b** Histological verification 4 weeks after the viral infection showing green fluorescence protein (GFP) in the neurons that positively express the construct. For anatomical reference, an image of the rat brain atlas is shown. Inset: 20× magnification picture of the same slice demonstrating selective infection of neurons in the NAc shell. **c**, 43**e**, **g**, **i** Single subject statistically thresholded fMRI maps showing voxels activated ($p < 0.001$, corrected) by perforant path stimulation and overlaid on an anatomical T2-weighted image. **d**, **f**, **h**, **j** BOLD time courses from significantly activated voxels averaged from the indicated regions of interests and across animals (mean ± SEM; $n = 6$ for **c**, $n = 3$ for **e**, **g**, and **i**). Details on fMRI processing and statistics are given in Supplementary Notes 2 and 8. **c** LTP experiment for comparison ((−) AAV infection, (−) CNO administration) showing the expected activation of HC, PFC, and NAc in POST-LTP. Note the evoked BOLD responses bilaterally in the HC (**d**), a landmark of HC response after LTP induction. **e** AAV infection in the NAc ((+) AAV, (−) CNO) preserves activation of the HC under perforant path stimulation before LTP. **g** Inactivating the NAc by administration of CNO in the same animal ((+) AAV, (+) CNO) does not alter functional maps nor BOLD responses in the baseline (PRE-LTP) condition (compare **e** vs. **g**). BOLD signal responses (**f**, **h**) are only evident in the ipsilateral HC as expected from PRE-LTP condition. **i**, **j** NAc inactivation ((+) AAV, (+) CNO) prevents the integration of the long-range network involving HC-PFC-NAc induced by LTP (POST-LTP)

**Network analysis of the resting-state dynamics.** As already indicated, the formation of the HC-PFC-NAc network is contingent on LTP induction. Accordingly, prior to LTP induction, the low-frequency stimulus that probes network function exclusively activates the HC, but neither PFC nor NAc are activated and, therefore, the relevance of these structures in the PRE-LTP condition cannot be studied during hippocampal stimulation.

To shed light on the role of these brain areas before LTP induction, we analyze resting-state fMRI data. From the fMRI signal prior to LTP, and in the absence of the low-frequency probing stimulus, we build a resting-state brain network for each of the six animals, by using the same network construction procedures as before. We then use CI centrality to rank the nodes according to their importance for brain integration, as we did for the LTP-induced functional network. Further details on the procedure are discussed in Supplementary Note 5 and an

averaged CI-map over the six rats is shown in Supplementary Fig. 4. These findings should be compared with Fig. 2h which presents the same type of results for the functional network induced by LTP.

The outcomes illustrate that, the NAc does not always play an essential integrative role. On the contrary, the importance of the NAc arises here as a result of LTP induction. In contrast, during resting state dynamics, nodes with high CI are distributed among different brain areas (see Supplementary Fig. 4). Therefore, the integrative role of the NAc is specifically related to synaptic plasticity in the memory network.

**Caveat on the methodology: from undirected to directed brain networks.** Key to our reasoning is that integrating information of specialized local modules into a global network is crucial for brain function. So far, this integration was modeled and measured as

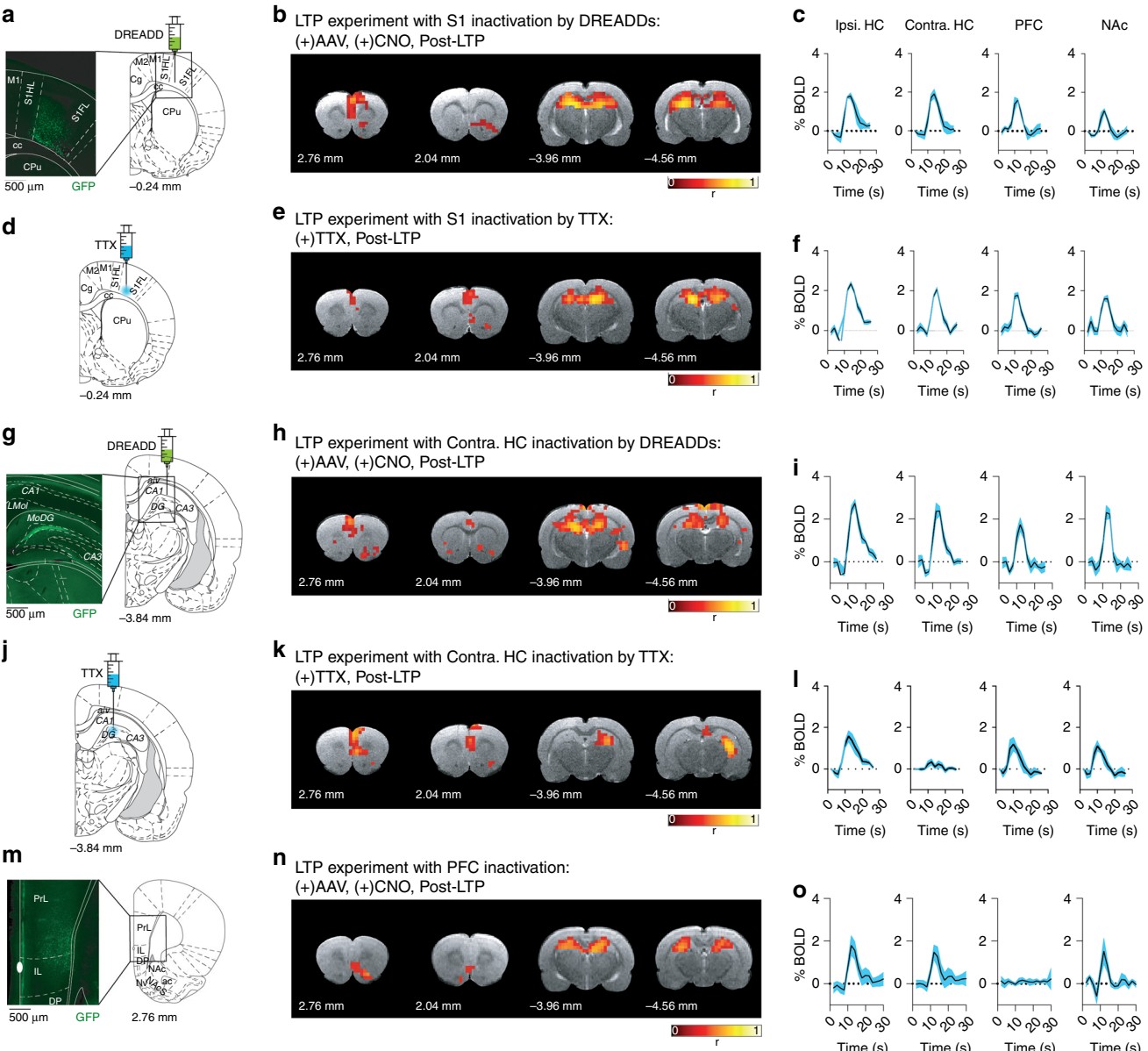

**Fig. 5** Targeted inactivation of different brain regions. **a–c** Pharmacogenetic inactivation of S1 cortex. **a** Location of the AAVs injection in the corresponding section of the stereotaxic map and representative histological staining showing the construct expression (inset). **b** Shows the statistically thresholded ($p <$ 0.001, corrected) fMRI maps of a representative animal and **c** the averaged BOLD signals across subjects ($n = 2$) and across region of interest. As in Fig. 4, S1 inactivation does not disrupt the long-range network formed upon LTP induction. **d–f** Inactivation of S1 with TTX (see Supplementary Note 9, for experimental details). Same as **a-b-c** experiments with S1 inactivation using the sodium channel blocker TTX ($n = 3$). Both fMRI maps and BOLD signals demonstrate formation of the HC-PFC-NAc network triggered by LTP ($p < 0.001$). **g–l** Pharmacogenetic (**g**, **h**, **i**, $n = 5$) and TTX (**j**, **k**, **l**, $n = 4$) inactivation in the contralateral HC ($n = 5$). As shown in the individual fMRI maps and averaged BOLD signals ($p < 0.001$), none of the inactivation strategies targeting the contralateral HC prevented the formation of the HC-PFC-NAc network. **m–o** Pharmacogenetic inactivation of the PFC ($n = 5$). AAVs injection targeted to the anterior part of the PFC (**m**) prevents its activation by performant path stimulation, as expected by the pharmacogenetic intervention, but does not abolish the formation of the long-range HC-NAc connections ($p < 0.001$), as predicted by the theory (**n**, **o**)

long-range correlated fMRI activity. However, these correlations do not necessarily measure direct interactions between neural populations through fibers, the so-called structural network. Some correlations may result from indirect covariations that do not reflect direct communication between nodes. To minimize effects due to this indirect covariations (i.e., high correlations between two nodes that are indirect since they do not come from a direct fiber structural connection between the nodes), we use a statistical approach (glasso)[38] which attempts to explain the

observed correlations as result of pairwise interactions. However, this model assumes undirected (symmetric) interactions. Measuring information exchange, on the other hand, needs a potentially asymmetric estimate that excludes some non-causal correlation, e.g., Granger Causality[47], which result in directed (asymmetric) interactions.

To determine if our results are robust when directed interactions are considered, we repeated the network analysis by endowing the network with directed links. For each pair of

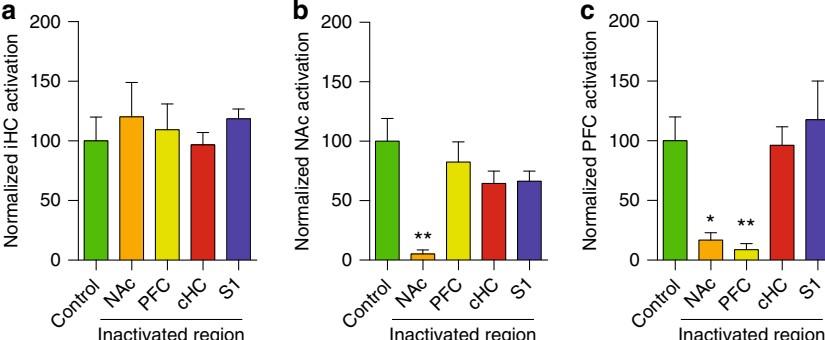

**Fig. 6** Group analysis of network inactivation. The number of nodes in the relevant networks is quantified after LTP induction with or without targeted inactivation and normalized relative to the control, fully active, condition. **a** Proportion of nodes recruited by perforant pathway stimulation in the ipsilateral HC under control conditions (100%) and after inactivation of the NAc, PFC, contralateral HC (cHC), and S1, as indicated in the x-axis. Analysis of variance across groups demonstrates no statistical differences (ANOVA, $F_{4,24} = 0.3641$, $p = 0.8317$). **b** Proportion of nodes recruited in the NAc following targeted inactivation in the structures indicated in the x-axis. ANOVA demonstrates statistically significant differences between groups ($F_{4,24} = 4.841$, $p = 0.0053$) and post-hoc Bonferroni test finds the only significant difference in NAc recruitment when the NAc is directly inactivated ($p < 0.01$), as expected from the experimental manipulation, but no effect under PFC, cHC, or S1 inactivation. **c** Proportion of nodes recruited in the PFC following targeted inactivation in the structures indicated in the x-axis. ANOVA demonstrates statistically significant differences between groups ($F_{4,24} = 6.416$, $p = 0.0012$) and post-hoc Bonferroni test identifies strong reductions in both PFC ($p < 0.01$), expected from the experimental manipulation, but also NAc ($p < 0.05$), indicating the disintegration of the long-range HC-PFC-NAc network under NAc inactivation

voxels in the HC-PFC-NAc network, we determined connectivity as before (Sec. 2) and, in addition, we measured Granger causality to determine the direction of the link. The final wiring of this directed network graph for each animal is different from the wiring of the undirected network (see Sec. 2). Remarkably, by computing the CI centrality on these directed networks (see Sec. 2 and Supplementary Note 6 for details), the main results regarding the location of the influential nodes is preserved: most influential nodes are located in the NAc and they are low-degree nodes, see Supplementary Fig. 5 in Supplementary Note 6. These results further strengthen our previous findings on the role of the NAc in the HC-PFC-NAc integration.

## Discussion

While a fundamental role of the NAc in the meso-cortico-limbic system has long been recognized, including for memory[48–50], our results suggest a new role for the NAc function in this system. The NAc receives major excitatory inputs from PFC and HC and dopaminergic inputs from the ventral tegmental area (VTA), among others[46,50]. These anatomical, but also neurophysiological and behavioral, evidences[49,50] have favored the view of the NAc as a downstream station in this circuit, working as a limbic-motor interface with a role in selecting behaviorally relevant actions[51]. Human and animal studies further indicate that in addition to performing on-line processing for action selection, the NAc encodes the output of the selected action (positive or negative relative to expectation) into memory, which in turn will condition future selections[49,50]. In this context, however, our network analysis locates the NAc upstream in the circuit, showing that interactions between the HC and PFC induced by LTP are already under the control of the NAc. Being the interaction between these two structures key for memory formation, we interpret our results as indicative of a NAc-operated gating mechanism that couples HC-PFC networks for the storage of new information, providing a mechanism for updating memories to guide future behaviors. This mechanism would fundamentally differ from, but being compatible with, previous ideas on information flow between HC, PFC, and NAc networks[52] in that the control here is exerted bottom-up from the NAc. While the precise mechanism for this control switch has not been investigated in the present work, an

appealing possibility is the regulation of neuronal excitability in the VTA by projections of the NAc shell through the ventral pallidum[48]. In turn, dopamine release from VTA terminals in the HC and neocortex would promote synaptic plasticity and facilitate integration in a consolidated memory brain network. Regardless of the specific microcircuit, in this network-driven theory, NAc computations seem to be a necessary part of hippocampal-dependent memories.

The experimental model used in this work leverages the induction of LTP in the dentate gyrus, which leads to a large-scale network that we could perturb prospectively. The experimental finding highlights the importance of considering the entire network associated with each node. Network hubs, defined solely by the number of direct connections, are not necessarily the most effective at channeling information through the entire network. This role may be reserved for essential nodes that connect different communities to each other[53]. The CI centrality used here accounts for the role of nodes in connecting different brain areas to one another[24]. Thus, this approach extends beyond the direct effects of hubs at integrating brain networks.

This result has important implications for the numerous investigations on brain pathology searching for critical alterations in functional connectivity as disease diagnostic and/or prognostic biomarkers. A combination of optimal percolation theory and experimental test presented here can be potentially adapted to networks that do not depend on LTP induction for their formation, thus providing a recipe to design intervention protocols to manipulate a wider range of brain states. These may include[9]: (i) transcranial magnetic stimulation that can stimulate or deactivate focal brain activity, (ii) assist in targeting deep brain stimulation devices, in particular, for disorders that are thought to be the result of network dysfunctions, and (iii) guiding brain tumor surgery by identifying essential areas to be avoided during the resection. The basic hypothesis is that activation/deactivation patterns applied to the influential nodes will propagate through the brain to impact global network dynamics. The proposed theoretical analysis provides a possible road map on how to establish and test such basic network hypotheses.

To conclude, we mention that our analysis was based only on correlation structure of evoked fMRI. Future work could study the

network structure and the role of node's degree in connectome data[54,55]. It would be important to compare the role of hubs, weak nodes, and nodes connecting different modules in structural brain networks with their role in functional networks. Such investigations, together with those presented in this work, are of crucial importance for diagnostic and clinical intervention in the brain.

**Data availability**. Data that support the findings of this study are publicly available and have been deposited in http://www.levich.engr.ccny.cuny.edu/webpage/hmakse/software-and-data/.

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

## Acknowledgements

This work was supported by NIH-NIBIB 1R01EB022720, NSF IIS-1515022, NIH-NCI U54CA137788/U54CA132378, NSF PHY-1305476, NIH-NINDS R01 NS095123, and by MINECO and FEDER Grants BFU2015-64380-C2-1-R, EU Horizon 2020 Grant No. 668863 (SyBil-AA), and Spanish State Research Agency, through the "Severo Ochoa" Program for Centers of Excellence in R&D (ref. SEV-2013-0317). Ú.P.-R. was supported by MECD Grant FPU13/03537. The authors are grateful to B. Fernández for technical assistance and K. Roth for discussions.

## Author contributions

All authors contributed to all parts of the study.

## Additional information

**Competing interests:** The authors declare no competing interests.

