## [Peer Review File · Nature Communications]

Reviewers' comments:

Reviewer #1 (Remarks to the Author):

Min et al. Finding influential nodes for integration in the brain using network optimization theory

The authors present an interesting study on combining LTP driven analysis with graph theoretical analysis in the rodent brain. The novelty of this paper is the use of a novel way to activate the network (by means of pharmaco) and then look for changes in network configuration.

My comments include:

I found the abstract well to read, but I nevertheless found it hard to directly figure out the main message. Perhaps less popular language and a little bit more actual results could make it more to the point, and better to understand for the novice reader where this study is about.

In their introduction the authors argue that a 'hub-centric' approach to select the most influential nodes might not always be the best way, as there is more than degree. Perhaps a slightly better presentation of the current literature might be in place here: there are several studies that have used other metrics that go beyond degree (eigenvector centrality, betweenness centrality etc).

the authors spend a lot of space to k-core decomposition in their introduction, but their point was not directly clear to me.

page 4: NoN was not introduced.

to be honest, I found the introduction quite long, with explanation of several centrality metrics which I found a little bit off topic. The real topic of investigation is not introduced until end of page 4

The term Network of Networks (NoN) is quite unclear to me. What do the authors mean by this?

The whole paper is full of abbreviations, which makes the text sometimes hard to follow

The first part of the results lacks statistical evaluation. The authors describe that they examined 6 rats and applied pre and post LTP induction, but no p-values are reported.

the expression 'After LTP induction,animals) was unclear to me. Do the authors mean that activity of the hippocampus was increased as the result of LTP? How do the authors conclude long-range NoN out of this?

Terms like L1, percolation theory, giant component etc are all not explained. This makes the paper rather hard to read for a novice reader.

The notion of node removal and the evaluation of this on network architecture has been well documented in literature, both in human and animal brain fMRI networks.

Throughout the paper many terms remain quite unclear. For example, what do the authors mean by 'hubs of the HC'. Are hubs only considered as voxels in the HC, and not throughout the brain?

The authors nicely consider network analysis during LTP. However, no control condition seem to be examined (i.e. the condition in which no LTP is examined). Without this null condition it is hard to see how these regions are essential for LTP (as claimed in the last sentence of page 11).

The nice bit of this paper comes when the authors introduce the pharmaco intervention of the

most central regions, which I greatly like. However, here too, I would have expected that the authors would have examined a few additional control regions, i.e. intervention in regions which their model predict to have no effect (i.e. non-central regions). Now, based on the current findings, it is not to distinguish whether these are true hub effects or just global effects.

The stimulation experiment lacks any statistical testing, I couldn't find any p-value or report of testing of effects in the results. Did the authors examine the lack of long-range connectivity in the simulated condition by means of some t-tests?

The conclusion of the authors to say that low-degree nodes are important for integration of brain networks (page 16) was unclear to me. Where did the authors report on the examined regions to be 'low degree'?

Studies have argued against the computation of 'functional hubs' based on fMRI data, as the degree of a node is influenced by the size of the network each node is in (Power et al. 2013). As a result, it is not surprising to see that degree (as based here on the functional data itself) is not directly related to inter-network connectivity (and therefore important to the giant core). The CI nodes are defined to interlink different modules, so their role in the giant component is not directly surprising. Others have thus argued that it might be better to define hubs based on anatomical data, not functional data. Can the authors respond on this, and report the critique on computing degree in functional networks? I didn't find the results presented in Figure 2 surprising, knowing this effect.

For the computation of the NoN, why did the authors only include nodes that were activated, and not whole brain? (i.e. all brain voxels)?

Reviewer #2 (Remarks to the Author):

This study combines analysis of fMRI data collected after an LTP protocol with intervention on network nodes to find nodes that are most influential for integration of information in the brain.

While the study is certainly conducted to a very high technical standard and its results have a very significant interest, there are several issues that need to be better addressed before being able to fully appreciate the importance of this study. These issues are listed below.

Key to the authors' reasoning is that integrating information of specialized localized networks is crucial for brain function, in particular it is essential to turn the brain into a coherent information processing system. In this study, this integration is measured and conceptualized exclusively as long range correlated fMRI activity. This view, stated at the very beginning of the introduction, is absolutely central to how the authors interpret every single result in their paper. I see dangers in this reasoning, none of them discussed in depth by the authors.

Is it sensible to assume that having highly correlated activity is good computationally? In the most correlated case, the brain (or a subnetwork) responds like a single node. Is thus looking for correlated activity a good way to look for intelligent computations or for interesting patterns of information flow?

Is it safe to argue that correlation between two nodes means that these two nodes exchange information? Would measuring information exchange need a potentially asymmetric measure that excludes some non-causal correlation as e.g. Granger Causality does?

The authors use correlation between fMRI activity, and these correlations likely include the effects of a lot of covariation factors that have nothing to do with real communication between nodes. In general, many studies both with fMRI and electrophysiology have shown that in many cases correlated activity and cross correlations reflect mostly covariations due to sources that do not

reflect functional, effective or anatomical connections between networks or neurons.

Most importantly, how do all these concerns affect the interpretation of the authors' results? What caveats do they force us to add? Which conclusions suffer mostly from these caveats and which conclusions suffer less?

The authors seem to concentrate these analysis on the networks created by LTP. I think that this analysis should be performed, possibly in the same subjects, also on spontaneously occurring networks in the unperturbed brain. Maybe the authors did this analysis but if so this is not emphasized strongly enough in the current version of the paper.

The authors keep referring to "in silico" studies that they did. I find this terminology very confusing. I expected to see studies done by actually implementing networks in silico (that is, in hardware). By reading the paper it seems that the "in silico" analyses are actually analyses where some nodes are selectively eliminated from the analysis. The authors may want to refer to these as analytical manipulations, or other similar terminology, but the term "in silico" should not be used in this context.

The analysis of the targeted inactivation data seems to be done on downsampled fMRI data (one voxel every 4). Was the downsampling also done on the analysis of the networks without targeted inactivation? If not why? Do the authors have a control of analysis of downsampled data collected without targeted inactivation? Any analysis that the authors did about this need to be better emphasized.

Reviewer #3 (Remarks to the Author):

In this paper, the authors explore the possibility to use network theoretical concepts, in particular centrality measures, in a neuronal setting, in order to identify regions central for brain integration. This work presents interesting contributions, in particular the empirical validation of the predictions by means of in vivo pharmacogenetic interventions. Unfortunately, it also suffers from important limitations, and I do not believe that it presents the technical novelty nor the broad scope justifying a publication in Nature Communications.

A first important drawback concerns the writing of the paper, with important stylistic flaws and, in particular, a lack of integration/cohesion between the different sections, in particular between the mathematical and neuronal parts of the work.

More importantly, the concept of NoN remains extremely vague throughout the paper and, even after reading the SI, the relevance and definition of the concept remains very unclear, as is its advantage over standard methods of network science, for instance based on community detection. The work also rests on a complicated methodology, involving several parameters and assumptions, raising questions about the generality of the results. The whole discussion on centrality measures, and observations like "the number of connections do not always lead to the most important nodes in the brain network", are fairly standard and basic in the field and I would suggest the authors to drastically reduce this section, in length and in claimed importance, before resubmitting their work to another journal.

Reply to Referee 1

“The authors present an interesting study on combining LTP driven analysis with graph theoretical analysis in the rodent brain. The novelty of this paper is the use of a novel way to activate the network (by means of pharmacology) and then look for changes in network configuration.”

We thank the referee for recognizing our study interesting and for considering our way to activate the network as a novelty. We reply below point by point to her/his comments.

Reviewer #1 (Remarks to authors):

1. “I found the abstract well to read, but I nevertheless found it hard to directly figure out the main message. Perhaps less popular language and a little bit more actual results could make it more to the point, and better to understand for the novice reader where this study is about.”

Following the Referee’s remark we have extensively rewritten the Abstract using less popular language and striving to go more directly to the point of our research. Below is the revised abstract:

“Global integration of information in the brain results from complex interactions of segregated brain networks. Identifying the most influential neuronal populations that efficiently bind these networks is a fundamental problem of systems neuroscience. Many network centrality measures have been proposed to identify these important areas in the brain, but few have been experimentally validated.

Here we apply network optimization theory and pharmacogenetic interventions in-vivo to predict and subsequently target nodes that are essential for global integration in a rodent model of learning and memory. We induce long-term potentiation of synapses in the hippocampus to establish functional long-range correlations with prefrontal neocortical regions and the nucleus accumbens. We then build a brain network model of these structures based on correlated activity recorded with functional magnetic resonance imaging. Using network theory we find the set of essential nodes that, once inactivated, will cause maximum damage to the largest connected component in the network. The method is based on the optimization of the collective influence rather than on the nodes’ number of local connections. The theory uniquely predicts that integration in this memory network is mediated by a set of low-degree nodes located in the nucleus accumbens. This theoretical prediction is confirmed with pharmacogenetic inactivation of the nucleus accumbens, which completely eliminates the formation of this memory network, while inactivation of other brain areas leave the network intact. Thus, network optimization theory can be used to predict essential nodes in functional brain network. This opens the way for targeted interventions that aim to manipulate the functional organization of the brain.”

2. “In their introduction the authors argue that a ‘hub-centric’ approach to select the most influential nodes might not always be the best way, as there is more than degree. Perhaps a slightly better presentation of the current literature might be in place here: there are several studies that have used other metrics that go beyond degree (eigenvector centrality, betweenness centrality etc).”

We thank the Referee for this remark since indeed there are various studies based on different centralities which tried to go beyond the ‘hub-centric approach’. We have added a full review of this literature in the revised manuscript with the following paragraph at end of Page 3 and beginning of Page 4, adding new references:

“ There are several studies that have used network centrality measures to identify the essential nodes in brain networks [3–6, 12, 17–20]. These measures includes the hubs (large-degree nodes,

i.e. nodes with many connections), betweenness centrality (BC) [21], closeness centrality (CC) [22], eigenvector centrality (EC) [23, 24], the k-core [25, 26], and collective influence centrality (CI) [27, 28]. A detailed definition of these centrality measures is provided in the Supplementary information (Section IV; see also [19, 29] for a review). ”

3. “the authors spend a lot of space to k-core decomposition in their introduction, but their point was not directly clear to me”

We agree with the Referee that our digression on k-core decomposition was excessively long. We have removed the discussion on k-core and just cited along other centralities that have been used to identify important nodes. Please see section ‘I. Introduction’ on pg. 3 of the revised MS.

4. “page 4: NoN was not introduced.”

We thank the Referee for spotting this omission. In the revised version of the MS, we have explained the concept of Network of Networks in terms of the more commonly used concept of ‘community detection’ in networks [41, 42] and introduced this notion in a clearer way in Section ‘II. B. Generate brain network model’. To further avoid confusion, we have also changed the term ‘Network of Network (NoN)’ with the more commonly used term ‘brain network’ in the neuroscience literature. Through the manuscript, we also sometimes refer to the same concept with the terms ‘brain network model’ or ‘HC-PFC-NAc network’, since this network is made by the wiring between the three anatomical regions which are the hippocampus (HC), the prefrontal cortex (PFC), and the nucleus accumbens (NAc). Please see also answer to point 6 below and answer to Referee 3, point 2.

5. “to be honest, I found the introduction quite long, with explanation of several centrality metrics which I found a little bit off topic. The real topic of investigation is not introduced until end of page 4”

We agree with the Referee that the Introduction, together with the report of the different centrality measure existing in the literature, was excessively long and also deviating from the main purpose of our article. Following the Referee’s advice we have considerably shorten and rewritten the Introduction, by moving the main topic of investigation - pharmacogenetic intervention in the rodent brain - to the beginning of our presentation. With this new arrangement we believe that the exposition gained in clarity and readability.

6. “The term Network of Networks (NoN) is quite unclear to me. What do the authors mean by this?”

We agree with the Referee that we have not being clear enough about this concept. We used the term Network of Network (NoN) to indicate the inferred functional brain network with the intent to emphasize its modular structure, i.e. an architecture which can be further described as the linkage of different brain regions (sub-networks). As mentioned in point 4 above, in the new version of the MS we replaced the nomenclature NoN with terms more commonly used in the literature. Furthermore, in Section II B ‘Generate brain network model’ we now explain in a clearer way the construction of this network, making explicit the connections with community detection algorithms, widely employed to identify the modular structure of large networks [41, 42].

7. “The whole paper is full of abbreviations, which makes the text sometimes hard to follow”

We agree with the Referee that we have used an excessive number of abbreviations. We have reduced them to a minimal amount and repeated, here and there in the text, what the acronyms stand for.

8. **“The first part of the results lacks statistical evaluation. The authors describe that they examined 6 rats and applied pre and post LTP induction, but no p-values are reported.”**

We thank the Referee for this significant remark and apologize that we have not made this point very clear. Statistical evaluations (p-value, F-values, etc.) for the different group comparisons, as well as for individual fMRI brain maps are reported in the captions of the corresponding figures (please see caption of Figure 1, 4, 5 and 6) and the corresponding statistical methods described in Supplementary Materials. For the sake of clarity now we have also reported some of them in the main text of the revised manuscript.

In addition, following the Referee’s remark, we have included additional pair-wise statistical comparisons between the different networks (hippocampus, prefrontal cortex, and nucleus accumbens) before (PRE-) and after (POST-) LTP induction (New Fig. 1E).

9. **“the expression ‘After LTP induction, ... animals’ was unclear to me. Do the authors mean that activity of the hippocampus was increased as the result of LTP? How do the authors conclude long-range NoN out of this? ”**

We thank the referee for this comment that allows us to clarify an important point in the work. Before LTP induction, stimulation of the perforant pathway - the main input to the hippocampal formation - produces the exclusive activation of the hippocampus. No activity is found neither in the prefrontal cortex nor in the accumbens during stimulation in the pre-LTP condition. In other words, the formation of a brain network composed by these three brain regions (i.e. HC + PFC + NAc) is contingent to the induction of LTP. Indeed this induction, by means of potentiating hippocampal synapses (Canals et. al. 2009, Ref. [35] in the manuscript), facilitates the formation of long-range functional connections and the appearance of a modular brain network.

To clarify this point we added a new figure, Fig. 1E, where we compare the fMRI activity of the three regions under study (HC, PFC, and NAc) without and with LTP stimulation (PRE and POST). As this new figure shows, LTP-induction enhances the activation of all the three brain regions. In particular, it increases the activation of the NAc and PFC, allowing for the formation of a three-modular brain network. Since the distances between different brain modules are long-range compared to distances within the same module, we conclude that LTP-stimulation induces long-range functional brain connectivity. This aspect has now been clarified in the text (page 7 at the end of section II. A) with the following text and reference to a new figure (Fig. 1E), where this result is reported:

“Compared to the baseline activation (PRE), LTP induction increases the evoked fMRI activity in a long-range functional network with enhanced bilateral fMRI activation of the HC, and activation in frontal and prefrontal neocortical regions (PFC), as well as the nucleus accumbens (NAc) ($p < 0.001$ corrected, see SI). Results are reported in Fig. 1D for a single animal, in Supplementary Fig. 7A for the average over six animals, and Fig. 1E quantifies group results and statistics. Conversely, low-frequency stimulation of the perforant pathway before LTP induction produces no fMRI activity neither in the prefrontal cortex nor in the nucleus accumbens.”

10. **“Terms like L1, percolation theory, giant component etc are all not explained. This makes the paper rather hard to read for a novice reader. ”**

We thank the Referee for this important comment and agree that a reader not expert in graph theory or machine learning may struggle to read this part of the paper. To improve our manuscript and its readability, we have removed many technical terms from the main text and explained better the meaning of the remaining ones which are ‘giant component’ and ‘percolation theory’. In pg. 9, Sec. II C we have changed the paragraph accordingly:

“We define global integration as the formation of the largest connected component of nodes in the network – the ‘giant connected component’ G (Fig. 2A), which is the component of the graph that contains a constant fraction of nodes in the limit of infinite system size. For finite systems, we consider the brain network globally integrated when its giant component is maximum, i.e. it contains all the nodes which are therefore all connected through a path. The emergence of such a giant component is an important concept in percolation theory [47, 48], a branch which studies the behaviour of clusters in networks by variation of some graph parameter.”

11. **“The notion of node removal and the evaluation of this on network architecture has been well documented in literature, both in human and animal brain fMRI networks”**

We thank the Referee for this remark. We added some reference to the node removal in fMRI brain networks and its effect on network architecture in neuroscience. In particular we added ref. [31–33] in section I. Introduction, Page 4, with the following paragraph:

“Some of these centrality measures have been studied using analytical and numerical manipulations and have been associated with different clinical phenotypes [5, 12, 19]. However, their importance for brain integration has not been directly tested experimentally with prospective interventions. Similarly, the notion of node removal and the evaluation of this on network architecture has been well documented in literature, both in human and animal brain fMRI networks [31–33], but direct in-vivo validations are rare. Thus, there is no well-grounded approach to predict which nodes are essential for brain integration.”

12. **“Throughout the paper many terms remain quite unclear. For example, what do the authors mean by ‘hubs of the HC’. Are hubs only considered as voxels in the HC, and not throughout the brain?”**

We took into account this and other Referee’s comments while attempting to improve the readability of the paper and clarify all the technical terms. By hubs we mean nodes (voxels) with high degree compared to the degree of the rest of the nodes (usually at least one order of magnitude higher). With ‘hubs of the HC’ we mean the hubs that are located in the hippocampus. We apologize for the lack of clarity. We now explain this point in the revised manuscript. We replace the original sentence, in Page 11 of the new MS, with:

“These results unveil a pattern in which centrality measures dominated by local degree (hubs, k -core, EC) tend to identify essential nodes in the hubs of the hippocampus, i.e. nodes with high degree are mostly located in the hippocampus region. These nodes, in the present experiment are trivially associated to the primary location of stimulation, while centrality measurements that capture long-range influence provide a non-trivial result highlighting the strength of the low-degree nodes at the NAc.”

13. **“The authors nicely consider network analysis during LTP. However, no control condition seem to be examined (i.e. the condition in which no LTP is examined). Without this null condition it is hard to see how these regions are essential for LTP (as claimed in the last sentence of page 11).”**

This is, again, an important issue raised by the reviewer that deserves clarification. As we explain in a previous point (point 9 above), stimulation of the HC in the pre-LTP condition does not recruit prefrontal cortex nor accumbens regions, thus, it does not activate a brain network made of different modules (please see the quantification of the fMRI activity shown in Fig. 1E with and without LTP stimulation). Therefore, the relevance of these structures in the pre-LTP condition cannot be studied during hippocampal stimulation. However, there is the possibility to study this condition using resting-state fMRI data.

To shed light on this question, we analysed resting-state fMRI data already acquired in our previous experiments and pertinent to a time window before LTP stimulation. For this new analysis we took into consideration the same anatomical areas studied to analyze the LTP case (same voxels), i.e. HC, PFC and NAc, this guarantees that nodes in the resting state network are the same as those of the LTP stimulation network.

From the fMRI signal of the brain, in this case, we repeated the whole analysis: we computed pair-wise correlations between time series, employed community detection and statistical inference techniques to construct the interacting network between the aforementioned brain areas and used the Collective Influence (CI) centrality measure to identify the integrators.

We present these results in a new section of the main text, Sec. II F “Network analysis of the Resting State dynamics”, pg. 16. Furthermore, we added additional technical details in a new section of the SI, Sec. VIII, where we also present a new figure, Fig. 10, which shows the CI-centrality map for the resting state dynamics, averaged over 6 animals.

Overall, results show that, the nucleus accumbens does not always play the role of brain integrator, regardless brain dynamics. On the contrary, this NAc’s crucial performance arises because of LTP stimulation. Indeed, when the brain undergoes different kind of dynamics, such as resting state dynamics, nodes with high CI are distributed among different brain areas (see Fig. 10 in the Supplement). Therefore, the nucleus accumbens significance for brain integration is highly related with its role in the LTP-induced memory network.

14. “The nice bit of this paper comes when the authors introduce the pharmaco intervention of the most central regions, which I greatly like. However, here too, I would have expected that the authors would have examined a few additional control regions, i.e. intervention in regions which their model predict to have no effect (i.e. non-central regions). Now, based on the current findings, it is not to distinguish whether these are true hub effects or just global effects.

We thank the referee for this important remark and believe that control experiments are a crucial validation of both our theory and previous findings. Moved by the Referee’s remark we have conducted a series of inactivation experiments targeting nodes that, based on our model predictions, should leave the functional network intact or largely preserved.

We recall that the brain network stimulated by LTP induction is made of three main regions: the hippocampus (HC), the prefrontal cortex (PFC) and the nucleus accumbens (NAc), see also Fig. 1D and 1F. Our theory predicts the NAc as the main area for integration and pharmaco-intervention proves that brain network formation is largely prevented by inactivating the NAc, as discussed in Sec. II C and D.

Therefore, as suggested by the Referee, we performed targeted inactivation in three additional targets. First, a control inactivation was targeted to the primary somatosensory cortex, a brain region located outside of the functional network considered in our study and which inactivation, as predicted by our theory, should have no effect. Two additional inactivations were targeted to structures in the HC-PFC-NAc network for which the CI centrality predicts mild or no effect in global integration. One of this inactivation was performed on the HC, contralateral to the stimulation, the other inactivation was performed on the anterior part of the PFC.

Each of these latter three inactivations had no effect on global integration and, as a result, the integrated HC-PFC-NAc network successfully formed. Overall, these results lend strong support to the predictive validity of our model and the key role of the NAc in the LTP-induced functional network.

Discussion and results are presented in the new section II E “Control experiments: *in-vivo* inactivation of brain regions predicted to have no effect”, pg 15. In the new Figures 5 and 6 we show the results of this comparison.

15. **“The stimulation experiment lacks any statistical testing, I couldn’t find any p-value or report of testing of effects in the results. Did the authors examine the lack of long-range connectivity in the simulated condition by means of some t-tests?”**

We thank the referee for the question, this remark has been already answered in point 8 above. In short, all functional maps represent statistical comparisons thresholded for p -values given in the caption of the figures. Group comparisons of the BOLD signals, when needed, were done based on ANOVA analysis. The same statistical tests for the new experiments have been added to the captions of the new figures. For the sake of clarity we have also reported some of them in the main text of the revised manuscript.

16. **“The conclusion of the authors to say that low-degree nodes are important for integration of brain networks (page 16) was unclear to me. Where did the authors report on the examined regions to be low degree?”**

We thank the referee for pointing out this lack of validation and apologize that we did not report on this point in the previous version of our MS. In order to address this remark, we have added a new section in the SI, Sec. X of the SI, together with a new figure, Fig. 12, where we show the degree statistics of the nodes in the rodent brain networks.

In particular, we show the degree distribution of the top 30 CI nodes of each animal, for all the animals, and compare it with the degree distribution of the top 30 hub nodes identified with a high-degree algorithm. We choose the first 30 nodes because, in most cases, after the first 30 CI top nodes are removed the network is completely dismantled. In other words, the list of CI nodes is almost never bigger than 30 nodes, across all animals. For completeness and clarity, in the same figure we also show the distribution of the whole brain network for all the animals. This figure illustrates that high CI nodes, i.e. nodes responsible for integration according to the collective influence centrality, are comparatively of lower degree than hubs in the brain network. Hubs are located mainly in the hippocampus and do not contribute considerably to the integration of the network.

In the main text we added the following paragraph towards the end of Sec. II C , Page 12, to further address the reader to these important result

“In Sec. X of the Supplement, we present the degree distribution of the CI nodes, across animals, and compare it with the distribution of the hubs. Fig. 12 in the Supplement illustrates that most of the top CI nodes are low-degree nodes.”

17. **“ Studies have argued against the computation of functional hubs based on fMRI data, as the degree of a node is influenced by the size of the network each node is in (Power et al. 2013). As a result, it is not surprising to see that degree (as based here on the functional data itself) is not directly related to inter-network connectivity (and therefor important to the giant core). The CI nodes are defined to interlink different modules, so there role in the giant component is not directly surprising. Others have thus argued that it might be better to define hubs based on anatomical data, not functional data. Can the authors respond on this, and report the critique on computing degree in functional networks? I didn’t find the results presented in Figure 2 surprising, knowing this effect.”**

We agree with the interpretation of the Referee that, for functional networks built from Pearson correlations, the node’s degree correlates with the size of the node’s community, as shown in Power et al. 2013. We thank the referee for pointing to this reference in the literature because it gives further understanding of the results we obtain through our theory.

Indeed, Power and collaborators show that functional hubs, defined uniquely through the node’s degree, do not play an important role for channeling information flow through a brain network.

They instead demonstrate that ‘articulation points’, regions in the brain connected with different brain communities, play a more important role. Their findings, as the Referee has pointed out, goes along the same lines as ours. Indeed, as noted by the Referee, the CI equation (2) gives higher score to nodes which connect different communities (second term on the RHS) and these nodes are not necessarily hubs. We therefore believe that our results - as well as those of Power and co. - point towards a re-thinking of the role and importance of hubs in brain networks. Moved by the Referee’s comment we have added the following paragraph towards the end of Sec. I. Introduction, Page 5, of the revised manuscript:

“More generally, the experimental confirmation of the crucial role of the low-degree integrator in the NAc shell suggests that the number of connections do not always lead to the most important nodes in the brain network. As shown by Power et. al [37], the degree of a node is influenced by the size of the network each node is in. As a consequence, functional hubs, defined uniquely through the node’s degree, do not always play the most important role for channeling information flow through the entire brain network. This essential role is primarily reserved to nodes that connect different communities [37] thus, it is not identified by the degree because this latter is not directly related to inter-network connectivity. The collective influence centrality that we use for our predictions accounts for nodes connecting different communities and, thus, goes beyond a hub-centric approach to identify integration in brain networks.”

Regarding the Referee’s comment **“Others have thus argued that it might be better to define hubs based on anatomical data, not functional data”** we believe that this is an important point. Further studies should focus on what is the role of hubs in anatomical networks, and compare their role to hubs in functional architectures. Yet, as it happens in functional networks (though for a different reason), also in anatomical brain architectures each node has many more connections with spatially close other nodes (in the same anatomical brain area / community) than with nodes located far apart (belonging to different anatomical brain area / community). Therefore we expect that the same phenomenon observed by Power et. al, which we also see with our theory, might arise in anatomical architectures. Indeed, nodes which connect different communities, so called articulation points, might have a more important role than degree-defined hubs for what concerns brain integration. Moved by the Referee’s remark we comment on this in the revised manuscript with the following paragraph, in Section III - “Concluding Remarks”.

“To conclude, we mention that our analysis was based only on correlation structure of evoked fMRI. Future work could study the network structure and the role of node’s degree in connectome data [57]. It would be important to compare the role of hubs, weak nodes, and nodes connecting different modules in structural brain networks with their role in functional networks. Such investigations, together with those presented in this work, are of crucial importance for diagnostic and clinical intervention in the brain.”

18. **“For the computation of the NoN, why did the authors only include nodes that were activated, and not whole brain? (i.e. all brain voxels)? ”**

We apologize for not stating our methods properly. We start by taking into account all brain voxels. Then we apply a statistical analysis to find the activated voxels. The activated voxels are those for which the BOLD signal has a statistically significant increase compared to a Resting State ($p < 0.001$) and therefore considered functionally active during stimulation. All other brain areas, not active during this process ($p \geq 0.001$) have to be considered as ‘functionally silent’, they do not have a statistically significant contribution to it, therefore are excluded from the brain network. We have clarified this analysis, stimulated by the Referee’s comment, and modified the text in the manuscript, Sec. II B, Page 7, as follows:

“The voxels with significant fMRI activation (due to the low-frequency probe after LTP induction, $p < 0.001$ corrected), form the nodes of the network model (see SI Section VI for details). The fMRI signal of the activated voxels is used to compute a functional connectivity matrix, i.e. pairwise correlations between voxels, separately for each animal.”

Reply to Referee 2

This study combines analysis of fMRI data collected after an LTP protocol with intervention on network nodes to find nodes that are most influential for integration of information in the brain.

While the study is certainly conducted to a very high technical standard and its results have a very significant interest, there are several issues that need to be better addressed before being able to fully appreciate the importance of this study. These issues are listed below.

We are grateful that the Referee considers our work conducted to a “very high technical standard” and of “significant interests”. We thank her/him for all the comments provided which we now fully address below.

Reviewer #2 (Remarks to authors):

1. “Key to the authors reasoning is that integrating information of specialized localized networks is crucial for brain function, in particular it is essential to turn the brain into a coherent information processing system. In this study, this integration is measured and conceptualized exclusively as long range correlated fMRI activity. This view, stated at the very beginning of the introduction, is absolutely central to how the authors interpret every single result in their paper. I see dangers in this reasoning, none of them discussed in depth by the authors.

Is it sensible to assume that having highly correlated activity is good computationally? In the most correlated case, the brain (or a subnetwork) responds like a single node. Is thus looking for correlated activity a good way to look for intelligent computations or for interesting patterns of information flow? Is it safe to argue that correlation between two nodes means that these two nodes exchange information? Would measuring information exchange need a potentially asymmetric measure that excludes some non-causal correlation as e.g. Granger Causality does?

The authors use correlation between fMRI activity, and these correlations likely include the effects of a lot of covariation factors that have nothing to do with real communication between nodes. In general, many studies both with fMRI and electrophysiology have shown that in many cases correlated activity and cross correlations reflect mostly covariations dues to sources that do not reflect functional, effective or anatomical connections between networks or neurons. Most importantly, how do all these concerns affect the interpretation of the authors results? What caveats do they force us to add? Which conclusions suffer mostly from these caveats and which conclusions suffer less?”

This is an extremely important point. In fact, we agree with the interpretation of the Referee that correlations between two nodes mostly reflect covariances and do not necessarily reflect the exchange of information. Instead, as suggested by the Referee, information exchange needs an asymmetric measure that excludes non-causal correlations such as, for instance, a Granger Causality analysis. We have addressed this important point in the manuscript three fold:

1. We first rewrite the Introduction to address this important point with the following paragraph:

“Integration in this context has been modelled and measured as correlated functional magnetic resonance imaging (fMRI) activity, often referred to as “functional connectivity” [2–6]. Correlations are not always expression of direct information flow though, indeed studies with both

fMRI and electrophysiology have shown that they may include effects of random covariations and therefore do not reflect functional, effective or anatomical connections between networks or neurons [7–9]. A more accurate measure of information exchange generally needs, indeed, a potentially asymmetric estimate that excludes some non-causal correlations, such as Granger causality [10]”

2. Second, we acknowledge the main concern of the referee as to **“which conclusions suffer mostly from these caveats”** by introducing the following considerations in a new Section II G - *Caveat on the methodology: from undirected to directed brain networks*, in pg. 17:

“Key to our reasoning is that integrating information of specialized local modules into a global network is crucial for brain function. So far, this integration was modelled and measured as long-range correlated fMRI activity. However, these correlations do not necessarily measure direct interactions, some of them may indeed contain effects of spurious covariations that do not reflect real communication between nodes. In fact, studies both with fMRI and electrophysiology have shown that correlations may reflect covariations due to sources that do not reflect functional, effective or anatomical connections. It is then important to understand how these assumptions affect the interpretation of the results, which conclusions are affected most and which conclusions suffer less.

To minimize effects due to spurious covariations, in our modelling we use a statistical approach [44] which attempts to explain the observed correlations as result of pairwise interactions. However, this model assumes undirected (symmetric) interactions. Measuring information exchange, on the other hand, needs a potentially asymmetric estimate that excludes some non-causal correlation, e.g. Granger Causality [10], which result in directed (asymmetric) interactions.”

3. Thirdly, we then address directly the concern of the Referee by performing again the whole network analysis but now adding a Granger causality measure to the links in the network. We follow this excellent suggestion and we have redone the whole network analysis using Granger causality to determine directed links between the nodes in the brain network. We follow standard procedures reviewed in the literature, see for instance E. Bullmore and O. Sporns, 2009, where it is reported:

“Most graph theoretical network studies to date have used symmetrical measures of statistical association or functional connectivity such as correlations, coherence and mutual information to construct undirected graphs. This approach could be generalized to consider asymmetrical measures of causal association or effective connectivity such as Granger causal [148,149] or dynamic causal [66] model coefficients, to construct directed graphs. It is also possible to avoid the thresholding step (step 3) by analysing weighted graphs that contain more information than the simpler unweighted and undirected graphs that have been the focus of attention to date. ”

66. Friston, K. J., Harrison, L., Penny, W. *Dynamic causal modelling. Neuroimage 19, 12731302 (2003)*

148. Roebroeck, A., Formisano, E., Goebel, R. *Mapping directed influence over the brain using Granger causality and fMRI. Neuroimage 25, 230242 (2005).*

149. Bressler, S. L., Tang, W., Sylvester, C., Shulman, G., Corbetta, M. *Top-down control of human visual cortex by frontal and parietal cortex in anticipatory visual spatial attention. J. Neurosci. 28, 1005610061 (2008).*

Moved by the Referee’s comment, we performed a network inference procedure accounting for Granger causality in order to exclude some non-causal relation among variables and, in addition, assign direction among the remaining ones. We performed the full network analysis for all the animals in our experiments by running a Granger causality test on all the time series (BOLD signals) selected for the initial analysis. For each pair of active voxels, not only we infer whether there is a connection linking them as before but, in addition, we run a Granger test to infer their causal relation and, therefore, the link directionality.

At the end of this procedure, for each rat, we end up with a directed brain network connecting the HC, the PFC and the NAc. In order to identify the integrators or influencers in this directed network we use a directed version of the CI algorithm which takes into account the link direction within the network, discussed in details in a new section of the SI, Sec. IX, Page 55.

The network that we obtain with the Granger test is different from the network obtained by the previous analysis based on undirected links. Remarkably, new results on directed networks indicate that the main conclusion is not affected: the shell in the nucleus accumbens remains the main essential node in the brain network. We therefore conclude that for the results of the present paper, a Granger causality analysis confirms the main prediction already found for undirected networks. A full new section is dedicated to this comparison, Section II G “*Caveat on the methodology: from undirected to directed brain networks*”, pg. 17. Further technical details are reported in a new section of the SI, Sec. IX, pg. 55. A new figure, Fig. 11, shows the CI-centrality map for the directed Granger causal networks, averaged over six animals, and should be compared with the same map for undirected networks shown in Fig. 2H.

We include the following paragraph in the main paper, in Page 18:

“To determine if our results are robust when probabilistic causal effects are considered, we repeated the network analysis by endowing the network with directed links. For each pair of voxels in the HC-PFC-NAc network, we infer whether there is a connection linking them as described in Sec. II B and, in addition, we run a Granger test to infer their causal relation, i.e. their link directionality. The final wiring of this directed network graph for each animal is different from the wiring of the undirected network (see Sec. II B). Remarkably, by computing the CI centrality on these directed networks (see Sec. II C and Supplementary Sec. IX for details), the main results regarding the location of the influential nodes is comparable: most influential nodes are located in the nucleus accumbens and they are low-degree nodes, see Fig. 11 in Sec. IX of the Supplementary Information. These results further strengthen our previous findings on the role of the NAc in the HC-PFC-NAc integration.”

We believe these results, moved by the Referee’s comment, to be very important to strengthen the role of the NAc in the rat brain even further.

Related to the following question moved by the Referee: “**In the most correlated case, the brain (or a subnetwork) responds like a single node. Is thus looking for correlated activity a good way to look for intelligent computations or for interesting patterns of information flow?**”, we fully agree that in the most correlated case, that is, when the brain is fully synchronized, all nodes respond as a single node, thus losing the capacity to transmit information. In fact, an excess of correlation could drive the brain into a complete synchronized state.

Thus, we have added a discussion to this point in Page 18 of the revised paper as follows:

“A related important problem is the response of the brain to an excess of external or internal inputs that could lead to the most correlated case, where the brain or a subnetwork of it responds like a single node. In this case, the brain behaves like a complete synchronised state in absence of external inputs with the loss of information transmission across different areas of the brain. The states that we find in the rat brain are not fully synchronized, instead they are balanced states of correlations between separated areas in the brain without reaching the fully synchronized state that would lead to a lack of information flows between activated areas.”

5. “The authors seem to concentrate these analysis on the networks created by LTP. I think that this analysis should be performed, possibly in the same subjects, also on spontaneously occurring networks in the unperturbed brain. Maybe the authors did this analysis but if so this is not emphasized strongly enough in the current version of the paper. ”

We thank the Referee for this important remark. Looking at the spontaneously occurring network in the unperturbed brain, the Resting State (RS) network, is indeed a way to better understand what is the role of each brain areas that we considered in our analysis. Indeed, in all our experiments, and in the new ones, we have measured the BOLD signal during RS of the unperturbed brain for the same animals, that is, before the LTP induction is applied, although we have not presented these results in the previous version of the MS.

In the revised manuscript we add the full network analysis of the unperturbed brain in a resting state condition, for all animals.

Results are reported in Section II G - “Network analysis of the Resting State dynamics” of the main text, pg. 18, with the following paragraphs:

“As already indicated, the formation of the HC-PFC-NAc network is contingent on LTP induction. Accordingly, prior to LTP induction, the low-frequency stimulus that probes network function, exclusively activates the HC, but neither PFC nor NAc are activated and, therefore, the relevance of these structures in the PRE-LTP condition cannot be studied during hippocampal stimulation.

To shed light on the role of these brain areas before LTP induction we analyze resting-state fMRI data. From the fMRI signal prior to LTP, and in the absence of the low-frequency probing stimulus, we build a resting-state brain network for each of the six animals, by using the same network construction procedures as before. We then use CI centrality to rank the nodes according to their importance for brain integration, as we did for the LTP-induced functional network. Further details on the procedure are discussed in Sec. IX and an averaged CI-map over the six rats is shown in Fig. 11 of the Supplementary information. These findings should be compared with Fig. 2H which presents the same type of results for the functional network induced by LTP.

The outcome illustrate that, the nucleus accumbens does not always play the role of brain integrator, regardless brain dynamics. On the contrary, this NAc’s crucial performance arises because of LTP stimulation. Indeed, when the brain undergoes different kind of dynamics, such as resting state dynamics, nodes with high CI are distributed among different brain areas (see Fig. 11 in the Supplement). Therefore, the nucleus accumbens significance for brain integration is highly related with its role in the LTP-induced memory network.”

6. “The authors keep referring to “in silico” studies that they did. I find this terminology very confusing. I expected to see studies done by actually implementing networks in silico (that is, in hardware). By reading the paper it seems that the “in silico” analyses are actually analyses where some nodes are selectively eliminated from the analysis. The authors may want to refer to these as analytical manipulations, or other similar terminology, but the term “in silico” should not be used in this context.”

We apologize for the wrong use of the term “*in silico*” and thank the Referee for spotting this misleading terminology. We have removed this nomenclature from the article and replaced it with either ‘numerically’, ‘numerical manipulations’, or ‘numerical predictions’ since the removal of nodes in the brain network is done by numerical simulations.

7. “The analysis of the targeted inactivation data seems to be done on downsampled fMRI data (one voxel every 4). Was the downsampling also done on the analysis of the networks without targeted inactivation? If not why? Do the authors have a control of analysis of downsampled data collected without targeted inactivation? Any analysis that the authors did about this need to be better emphasized.”

We thank the Referee for this comment, we realize that this point was not very clear in our previous manuscript. All the analysis that we performed (also the new ones on resting state) are all done on downsampled fMRI data, same downsampling for all the data. The reason to work with downsample data is twofold: (a) The original BOLD signal is measured at a resolution of $0.26 \times 0.26 \times 1$ mm in the three spatial dimensions. This resolution creates an anisotropic voxel which in turn creates anisotropic correlations after the signal is coregistered in the Paxinos atlas for further processing. Corregistration into a common atlas is the established procedure in neuroscience to compare different rats to obtain average quantities across animals. It is a requirement of the corregistration process that the voxels needs to be isotropic. (b) To avoid spurious correlations in the fMRI transverse plane ($x - y$), introduced by the Gaussian kernel spatial smoothing procedure, we consider only one voxel every four in these directions. This downsampling produces a voxel volume of $1.04 \times 1.04 \times 1$ mm, which is roughly isotropic and of the same size of the target pharmacogenetic interventions and produce isotropic correlations between the voxels. We comment on this adding the above paragraph in Sec. II D, at the end pg. 13, of the revised manuscript.

*“In order to avoid spurious correlations between neighboring nodes in the fMRI transverse plane introduced by the Gaussian kernel spatial smoothing procedure applied to the image (Supplementary information Section VD), we consider a node in the network only every fourth voxels in the x and y directions. Thus, the nodes in the **brain network model** are separated by $1.04 \times 1.04 \times 1$ mm, are approximately isotropic in all three dimensions, and represent approximately 1mm^3 of volume. The same downsampling procedure is applied in all the analysis described in the text, with or without pharmacogenetic intervention, see SI, Sec VI B for further details”*

Furthermore, we have explained more clearly this procedure in the SI, at the beginning of Sec. VI. B, Page 48, as mentioned in the MS with the text reported above, with the following paragraph:

*“Correlations are computed separately for each animal for all voxels that showed significant activation in at least 2 animals (activation maps were co-registered to a standard atlas, but correlation is computed in the original space to avoid introducing spurious correlations due to resampling). In the animal original space, the BOLD signal is measured at a resolution of $0.26 \times 0.26 \times 1$ mm. Another source of spurious correlations might arise when applying Gaussian kernel spatial smoothing procedure applied to the image, because the volume space is not isotropic. So, to avoid including spurious correlations in the fMRI (x, y) plan, we consider only every fourth voxels so that nodes are separated by $1.04 \times 1.04 \times 1$ mm, and are approximately isotropic in all three dimensions. Therefore, the size of the voxel, that is, each node in the **brain network**, is approximately 1mm^3 and this corresponds to a single node in the network. This size is commensurate with the size of the target in the pharmacogenetic interventions. The same downsampling procedure described above is applied in all the analysis described in the text, with or without pharmacogenetic intervention. ”*

Reply to Referee 3

Reviewer #3 (Remarks to authors):

“In this paper, the authors explore the possibility to use network theoretical concepts, in particular centrality measures, in a neuronal setting, in order to identify regions central for brain integration. This work presents interesting contributions, in particular the empirical validation of the predictions by means of in vivo pharmacogenetic interventions. Unfortunately, it also suffers from important limitations, and I do not believe that it presents the technical novelty nor the broad scope justifying a publication in Nature Communications.”

We are glad that the Referee thinks our work presents interesting contribution, in particular the empirical validation of the predictions by means of in vivo pharmacogenetic interventions. Below we address all Referee’s critiques. The manuscript has been overall largely rewritten, new numerical and *in-vivo* experimental validations of our theory (as suggested by Referee 1 and 2) have been conducted and the new findings further strengthen our previous results. To address all the Referees’ comments and report all the new experimental and numerical validations, new figures and sections have been added to the manuscript.

In details, Sec I. - Introduction and Sec. II. B have been largely rewritten. Three new sections (Sec. II. - E, F, and G) have been added to report the findings of the new *in-vivo* experiments and of the new numerical analyses. In the Supplementary Information, Sec. VI. B and C have been largely rewritten and four new sections have been added, Sec. VIII, IX, X and XII.

New figures have been included in the main text - Fig. 1E, Fig. 5 and 6 - to support the new findings and address the Referees’ comments. In the Supplementary Information the new Fig. 8, 10, 11 and 12 have been included for further details.

We hope that, after reading the new version of the MS, Referee 3 will reconsider our paper for publication.

1. “A first important drawback concerns the writing of the paper, with important stylistic flaws and, in particular, a lack of integration/cohesion between the different sections, in particular between the mathematical and neuronal parts of the work.”

Thanks to all the Referees’ comments the manuscript has been largely rewritten, the order of some concepts changed and, overall, we have attempted to produce a more cohesive text with the different sections (we believe) better integrated among each other. Joining mathematical and neuroscience efforts, descriptions and analysis, is not always straightforward and we agree with the Referee that our previous version of the MS resulted in some part ‘disconnected’, not fluent or with unbalanced flow among the different sections. This point was also noticed by Referee 1 and 2, who suggested to remove the discussions in the Introduction on different centralities and explain directly the experimental and neuro-related results. Because of the new experiments and findings, and also moved by Referee 3’s comment, we have made major revisions to our manuscript by largely rewriting few sections and adding new ones. We hope that our work is now of broader importance, of easier reading and more suitable for publication.

Following Referee 3, 1 and 2’s comments we have rewritten the whole Introduction. Please see text starting on pg. 3 of the revised MS.

2. “More importantly, the concept of NoN remains extremely vague throughout the paper and, even after reading the SI, the relevance and definition of the concept

remains very unclear, as is its advantage over standard methods of network science, for instance based on community detection”

We truly apologize for not explaining correctly the concept of NoN. Indeed this concept refers to a network with a community structure (connected sub-networks identifiable as different communities). Thus, the term NoN refers to a network build through “community detection” procedures, a notion well known in network literature and which has been widely applied to complex networks with, for instance, algorithms as the Girvan-Newman’s [41, 42], as mentioned by the referee. We have called it NoN in our original manuscript following certain literature in the field of network theory, but we now recognize the need to make clear the connection between our method and community detection algorithms, which are more standard in the literature. For this reason, section II B ‘Generate brain network model’, Page 7, has been completely rewritten to explain our method as a community detection procedure, which is at the basis of our brain network definition. The term NoN has been removed from the manuscript and replaced by ‘brain network’ or ‘HC-PFC-NAc network’, to highlight its community structure as a network which connects the different brain modules: hippocampus (HC), prefrontal cortex (PFC), and nucleus accumbens (NAc).

Our method has been developed in the neuroscience literature as well as in the network literature. It is routinely used in a number of works, for instance, a well known review in the field, Bullmore, E. & Sporns, O. ‘Complex brain networks: graph theoretical analysis of structural and functional systems’. *Nature Rev. Neurosci.* **10**, 186-198 (2009), which has 5400 Google Scholar citations.

Our network construction is alike to the method depicted in Fig. 1 of the aforementioned review (see Figure 1 below) to construct a brain network from BOLD time series. We quote from this paper

“Such networks are derived from measures of association between the simulated time series for example, an information theoretic measure such as the mutual information (computed on voltage-time data) or cross-correlations in neural activity that are computed from simulated blood oxygen level-dependent (BOLD) data. These matrices can then be thresholded to yield binary networks from which network measures can be derived.”

Following standard literature, as the method described by Bullmore and Sporns in their 2009 review, the starting point of our analysis is the cross-correlation matrix, obtained from the BOLD signal of all the active voxels in the rodent brain. On this matrix we apply a community detection algorithm to identify the clusters that corresponds to the anatomical areas of the hippocampus, the nucleus accumbens and the pre-frontal cortex.

The network architecture can be obtained from this cross-correlation matrix by thresholding the elements and so building an adjacency matrix. Again, from Bullmore and Sporns, 2009, already quoted above *“These matrices can then be thresholded to yield binary networks from which network measures can be derived.”* Or similarly, from Box 1 of the same review *“Step 3. Generate an association matrix by compiling all pairwise associations between nodes and (usually) apply a threshold to each element of this matrix to produce a binary adjacency matrix or undirected graph”.*

The issue of thresholding mentioned in this review, i.e. the value at which to fix the threshold, is solved in the literature in several ways, depending on the features of the network that one is interested to reconstruct, e.g. robustness, wiring efficiency, sparsity, etc... From the review quoted above:

“A crucial issue at step 3 is the choice of threshold used to generate an adjacency matrix from the association matrix: different thresholds will generate graphs of different sparsity or connection density, and so network properties are often explored over a range of plausible thresholds”

In our work, we do not directly build the network by thresholding the cross-correlation matrix but, rather, we make one step forward to improve the final result. Indeed, as pointed by the Referee 2, building networks by thresholding the cross-correlation matrix might add fictitious links in the final network due to covariation factors that have nothing to do with real communication between nodes. In addition, correlations are obtained from a finite sampling of time series (the fMRI BOLD signal) and not dealing with infinite time series already brings some source of error in the data set. To deal with this two issues - *(i)*, fictitious connections due to covariations and *(ii)*, finite size sampling of the time series - and finally infer the network architecture, we employ a very well established statistical inference method by Tibshirani and collaborators (which has more than 22.000 Google Scholar citations). This approach, named Least Absolute Shrinkage and Selection Operator (LASSO), infers a sparse representation of the network and reduces the noise due to finite size sampling.

Tibshirani, R. "Regression shrinkage and selection via the lasso." Journal of the Royal Statistical Society. Series B (Methodological) (1996): 267-288.

Friedman, J., Hastie, T. and Tibshirani, R. "Sparse inverse covariance estimation with the graphical lasso." Biostatistics 9.3 (2008): 432-441.

The LASSO approach does not eliminate the issue of thresholding, indeed the network sparsity is tuned by a parameter which acts like a threshold. As it happens for the correlation thresholding, also here *"different thresholds will generate graphs of different sparsity or connection density"*, quoting Sporns and Bullmore above. We are interested in the appearance/disappearance of the giant component in the network, since it is the topological property that we use to identify brain network integration, thus we fix the value of the threshold to the highest value at which a giant component is maximum. In other words, we fix the threshold such that, the final network is the sparsest possible architecture which includes all the nodes. A sparse network guarantees that at least some of the fictitious connections mentioned above from thresholding correlations are eliminated, whereas a giant component assures that each node is connected through a path with any other node and so brain integration (as we model it) is, in principle, possible.

Summarizing, from the cross-correlation matrix we employ: *(i)*, community detection to detect the node's community and *(ii)*, LASSO to infer a sparse representation of the brain network which reduces: errors due to covariation not related with real communication between nodes and errors due to finite size sampling. The threshold parameter is chosen to guarantee a giant component in a systematic way (reproducible results) and based on a physical argument (the giant component is the parameter in our theory related to brain integration).

We have largely rewritten Sec. II B "Generate brain network model", in order to explain this procedure more clearly and make the connection with community detection explicit. Technical details are reported in the SI, Sec. VI, which has also been largely rewritten. In Fig. 8 of the SI we show the resulting adjacency matrix of the brain network for a representative animal. This matrix shows a clear modular structure organization of the brain, typical of graphs with network communities, similar to the matrices shown in Fig. 1 of the review by Ed. Bullmore and O. Sporns (2009), which is included in at the end of this reply for clearness.

As additional test of our theory and its prediction, following the suggestion of Referee 2, we perform a supplementary analysis. We apply a Granger causality test to obtain an asymmetric directed graph. Discussion and new results regarding the effect of topological asymmetries in the brain network are reported in the new Sec. II G of the main text. This procedure is also standard and has been reviewed in the aforementioned review by Bullmore, E. and Sporns, O. from which we quote:

“Most graph theoretical network studies to date have used symmetrical measures of statistical association or functional connectivity such as correlations, coherence and mutual information to construct undirected graphs. This approach could be generalized to consider asymmetrical measures of causal association or effective connectivity such as Granger causal [148,149] or dynamic causal [66] model coefficients to construct directed graphs.”

[66] Friston, K. J., Harrison, L. & Penny, W. *Dynamic causal modelling. Neuroimage 19, 1273-1302 (2003)*

[148] Roebroeck, A., Formisano, E. & Goebel, R. *Mapping directed influence over the brain using Granger causality and fMRI. Neuroimage 25, 230242 (2005).*

[149] Bressler, S. L., Tang, W., Sylvester, C., Shulman, G. & Corbetta, M. *Top-down control of human visual cortex by frontal and parietal cortex in anticipatory visual spatial attention. J. Neurosci. 28, 1005610061 (2008).*

We provide additional results on directed architectures in a new section of the SI, Sec. IX, where, with the new Fig. 11, we present the average prediction of nodes responsible for integration in directed brain networks. Remarkably, also in directed networks, the majority of the influential nodes are located in the nucleus accumbens.

Furthermore, as suggested by Referee 1, we performed also a network analysis of the influential nodes for integration for the unperturbed brain, during resting state dynamics. These new results are discussed in Sec. II F and further details are reported in the Supplement, Sec. VIII.

To conclude, the Referee’s comment on the relation between our procedure and standard community detection methods to identify the modules is very pertinent and interesting. In principle it is indeed possible to apply any sort of community detection algorithm to identify the communities (clusters).

We here stress, though, that our major contribution in this work is not in the construction of the NoN (or brain network) which could indeed be made by using other algorithmic methods, as pointed by the Referee. Our main research finding is the identification of the areas driving the integration in the brain and the pharmaco-intervention validation *in vivo* of our theoretical predictions.

In this respect, as asked by Referee 1 and 2 we have performed further experimental validations of our theoretical predictions. In the previous version of the MS, we performed pharmaco-genetic inactivation only of the nucleus accumbens (NAc), which is the region predicted to be responsible for brain integration by our theory.

To further validate this prediction we performed additional pharmaco-genetic interventions on three more brain areas. One targeted intervention was performed on the primary somatosensory cortex, a brain area located outside the functional network stimulated by LTP, made of the HC, the PFC and the NAc. Two additional targeted inactivations were made on regions of the LTP network on which our theory predicts to have mild or no effect for global integration, i.e. the HC and the PFC. Please see also answer to Referee 1, point 14.

These new findings are presented in Sec. II. E ‘Control experiments: *in-vivo* inactivation of brain regions predicted to have no effect’ of the main text. Overall, the new experiments and numerical tests further strengthen our previous findings which identify the NAc as the main actor for brain integration in the LTP-induced functional network.

3. “The work also rests on a complicated methodology, involving several parameters and assumptions, raising questions about the generality of the results.

We apologize again for not explaining the generality of our approach. We have now removed the concept of NoN from the manuscript considering that it does not explain well the methods used in

Figure 1 | Computational modelling of structural and functional brain networks. Computational models have been used to demonstrate how dynamic patterns arise as a result of interactions between anatomically connected neural units. Shown is how such a model is generated and used. A structural brain network derived from anatomical data serves as a matrix of coupling coefficients that link neuronal nodes, the activities of which unfold through time. This time evolution is governed by physiologically motivated dynamic equations. In the example shown, the surface of the macaque cortex was subdivided into 47 areas (nodes) and a structural brain network linking these nodes was compiled from anatomical tract-tracing data. The dynamic equations were derived from a model of large neuronal populations, the parameters of which were set to physiological values¹⁰⁹. Data from computer simulations then yield functional brain networks. Such networks are derived from measures of association between the simulated time series — for example, an information theoretic measure such as the mutual information (computed on voltage–time data) or cross-correlations in neural activity that are computed from simulated blood oxygen level-dependent (BOLD) data. These matrices can then be thresholded to yield binary networks from which network measures can be derived. The fact that both structural and functional networks are completely specified in the model facilitates their comparative analysis. The structural brain network panel is reproduced, with permission, from REF. 109 © (2007) National Academy of Sciences. The rest of the figure is modified, with permission, from REF. 158 © (2009) Academic Press.

FIG. 1: From Ed. Bullmore and O. Sporns (2009). Our method to construct the brain network starts from this covariance matrix on which we apply a community detection algorithm to identify the network communities and a statistical inference approach to obtain a sparse representation of the brain network, as explained in our revised manuscript.

the computational neuroscience and network literature. Since our procedure is based on community detection mentioned by the Referee and used and reviewed in many papers, see for instance the review by Bullmore, E. and Sporns, O. (2009) above, we now explain our methodology in terms of community detection and statistical inference as suggested by the Referee. The entire paper is now formulated in terms of these two approaches which makes it also general enough since it is based on common methods used in the literature (as the Referee pointed out). We hope that these modifications may answer the questions on generality put forward by the Referee, well founded in the fact that the NoN is not standard terminology in the network literature.

4. “The whole discussion on centrality measures, and observations like “the number of connections do not always lead to the most important nodes in the brain network”, are fairly standard and basic in the field and I would suggest the authors to drastically reduce this section, in length and in claimed importance, before resubmitting their work to another journal.”

We agree with the referee that the discussion on centrality measure is fairly standard and basic in the literature and that, therefore, can be reduced in length. We have removed the sentence mentioned by the referee, the whole discussion on centrality measures has been shorten and the pertinent literature has been cited, instead. We have drastically reduced this sections in length and in claimed importance. We have now rewritten this part and replaced it with the following paragraph in pg. 3-4.

“There are several studies that have used network centrality measures to identify the essential nodes in brain networks [3–6, 12, 17–20]. These measures includes the hubs (large-degree nodes, i.e. nodes with many connections), betweenness centrality (BC) [21], closeness centrality (CC) [22], eigenvector centrality (EC) [23, 24], the k-core [25, 26], and collective influence centrality (CI) [27, 28]. A detailed definition of these centrality measures is provided in the Supplementary information (Section IV; see also [19, 29] for a review).”

Our hope is that the modifications done by following the Referee’s suggestions have strengthen the conclusions such that the paper can be reconsidered for publication.

REVIEWERS' COMMENTS:

Reviewer #2 (Remarks to the Author):

I am fully satisfied by the revisions.

Reviewer #3 (Remarks to the Author):

The form of this manuscript has been greatly improved since the original submission. It allows to have a more clear idea of the objectives and methodologies developed in the paper. As the authors emphasise in their rebuttal letter, the whole concept of NoN is not necessary, and the computational methodology used by the authors to uncover important nodes is actually relatively standard. My understanding is now that the main contribution of this paper would be the use of existing methods to uncover integrating regions of the brain, which has been studied before in the literature, and a test of the computational predictions by in-vivo interventions.

Despite its improvement, the draft still requires some clarifications on the methodological side. The authors now argue that the brain is organised on a modular way, and I would have expected a better review of existing works along those lines, for instance starting from the review: Modular and hierarchically modular organization of brain networks, David Meunier et al. Moreover, it is surprising that the authors do not exploit information from the modular organisation in order to predict important nodes. For instance, when they write: "Connections between nodes belonging to different clusters are named inter-links, or weak-links, reflecting the long-range interactions between different sub-networks.", this echoes the works of Functional cartography of complex metabolic networks, Guimera et al. about functional roles in terms of their location at the core or at the border of communities. The authors should also provide a more thorough literature review on works combining networks science and interventions to convince of the novelty of their work. See for instance the recent: Network Control Principles Predict Neuron Function in the *Caenorhabditis elegans* Connectome, Yan et al.

Response to Reviewers' comments:

We are really glad that we fully satisfied Referee 2 with our revisions.

We are also pleased that Referee 3 finds our work “greatly improved since the original submission”. We thank him/her for pointing to some solid paper in the literature that should be cited along the text. We added these references which are Ref. 34, 41, and 56 in the reviewed version of the manuscript.